# Linc00673-V3 positively regulates autophagy by promoting Smad3-mediated *LC3B* transcription in NSCLC

Heng Ni[1,2,*], Song Tang[1,*], Guang Lu[3,*] , Yuequn Niu[1,2], Jinming Xu[2], Honghe Zhang[4], Jian Hu[2], Han-Ming Shen[5,6], Yihua Wu[1] , Dajing Xia[1] 

**Since its first discovery, long noncoding RNA Linc00673 has been linked to carcinogenesis and metastasis of various human cancers. Linc00673 had five transcriptional isoforms and their biological functions remained to be explored. Here we have reported that Linc00673-V3, one of the isoforms of Linc00673, promoted non–small cell lung cancer chemoresistance, and increased Linc00673-V3 expression level was associated with enhanced autophagy. Mechanistically, we discerned the existence of a stem-loop configuration engendered by the 1–100-nt and 2200–2275-nt fragments within Linc00673-V3. This structure inherently interacted with Smad3, thereby impeding its ubiquitination and subsequent degradation orchestrated by E3 ligase STUB1. The accumulation of Smad3 contributed to autophagy via up-regulation of *LC3B* transcription and ultimately conferred chemoresistance in NSCLC. Our results revealed a novel transcriptional regulation network between Linc00673-V3, Smad3, and *LC3B*, which provided an important insight into the interplay between autophagy regulation and non-canonical function of Smad3. Furthermore, the results from in vivo experiments suggested Linc00673-V3 targeted antisense oligonucleotide as a promising therapeutic strategy to overcome chemotherapy resistance in NSCLC.**

## Introduction

Lung cancer is the second most prevalent cancer and the leading cause of cancer mortality worldwide, with an estimated 2.2 million new cases and 1.8 million new deaths in 2020 (Sung et al, 2021). Despite the emergence of novel targeted therapy and immunotherapy, cisplatin-based chemotherapy is still the first-line treatment for advanced NSCLC patients. However, resistance to cisplatin-based chemotherapy occurs frequently and it remains the most significant obstacle in terms of improving the long-term outcomes of patients with advanced stages (Chang, 2011). The molecular mechanisms underlying NSCLC cisplatin resistance are multifactorial and not fully understood, with increasing studies shed light on the emerging roles of autophagy in NSCLC cisplatin resistance (Wu et al, 2015; Bai et al, 2019; Ma et al, 2019; Pan et al, 2019; Limagne et al, 2022).

Autophagy is an evolutionarily conserved catabolic process involving the degradation of cellular contents to allow basal turnover of cellular components to provide macromolecular precursors as well as energy, which supports cell survival under stress conditions (Mizushima, 2007; Kim & Lee, 2014; Lu et al, 2021). A series of evolutionarily conserved autophagy-related (ATG) genes have been discovered and among these ATG proteins, *ATG8* was found essential in cargo recognition, engulfment, and vesicle closure in yeast (Levine & Kroemer, 2019; Lu et al, 2022). In higher eukaryotes, *ATG8* has evolved into *GABARAP* (GABA type A receptor–associated protein) and *MAP1LC3* (microtubule associated protein 1 light chain 3) subfamily, the latter is often referred to as *LC3*. LC3 protein, a well-established marker of autophagosomes, exists in two main forms: LC3I and LC3II. LC3I, the cytosolic form, is lipidated to form LC3II through a series of sequential steps during autophagy initiation. The conversion of LC3I to LC3II involves the conjugation of phosphatidylethanolamine (PE) to LC3I, which is mediated by the ATG protein complex. LC3II is then recruited to the growing autophagosomal membrane, where it plays a pivotal role in cargo recognition and autophagosome maturation (Jacquet et al, 2021). The role of autophagy in cancer is context dependent. Although it is documented that autophagy hinders tumorigenesis by facilitating the elimination of cytotoxic cytosolic contents, at advanced stages, heightened autophagic flux empowers tumor cells to endure adverse conditions, thereby conferring resistance to chemotherapeutic stress (White, 2012; Levy et al, 2017; Hanahan, 2022). Based on these theories, clinical trials have been carried out in opposite ways

[1]Department of Toxicology of School of Public Health and Department of Gynecologic Oncology of Women's Hospital, Zhejiang University School of Medicine, Hangzhou, China   [2]Department of Thoracic Surgery, First Affiliated Hospital, School of Medicine, Zhejiang University, Hangzhou, China   [3]Zhongshan School of Medicine, Sun Yat-sen University, Guangzhou, China   [4]Department of Pathology and Women's Hospital, Zhejiang University School of Medicine, Hangzhou, China   [5]Department of Physiology, Yong Loo Lin School of Medicine, National University of Singapore, Singapore, Singapore   [6]Faculty of Health Sciences, Ministry of Education Frontiers Science Center for Precision Oncology, University of Macau, Macau, China

Correspondence: georgewu@zju.edu.cn; dxia@zju.edu.cn
*Heng Ni, Song Tang, and Guang Lu contributed equally to this work

to interfere autophagy in lung cancer, but the results are still controversial (Malhotra et al, 2019; Vidal et al, 2021).

LncRNA refers to those >200-nt ncRNA, with no evident open reading frames nor coding capacity. It is reported that lncRNA participates in regulation of autophagy by modulating the promoter regions of ATG genes (Wang et al, 2019), stabilizing cytoplasmic mRNA of ATG genes (YiRen et al, 2017; Zhou et al, 2020; Luo et al, 2021) and protecting translation products of ATG genes from ubiquitination and degradation (Cai et al, 2019). Smad3 is well known for its transcriptional role in TGF-β signaling pathway; however, recent studies have uncovered its potential role in autophagy. Yang et al (2021) reported that Smad3 inhibited TFEB-dependent lysosome biogenesis which leads to autophagy dysregulation in diabetic nephropathy. On the other hand, Shim et al found that depletion of Smad2/3 significantly decreased the protein level of LC3II, whereas non-canonical Smad2/3 signaling is critical to maintain autophagy in trabecular meshwork cells (Shim et al, 2021). Further investigation is warranted to comprehensively elucidate the precise roles and underlying mechanisms of Smad3 in the regulation of autophagy.

In this study, we elucidate a pivotal regulatory function of Linc00673-V3 in autophagy within NSCLC cells. Our findings demonstrate that Linc00673-V3 plays a critical role in stabilizing Smad3, consequently promoting elevated transcription of *LC3B* by Smad3, and thereby facilitating enhanced autophagy. Importantly, this heightened autophagic activity contributes to the development of chemoresistance in NSCLC. Taken together, these results suggest Linc00673-V3 as a promising therapeutic target for combating chemoresistance in NSCLC patients.

# Results

### Linc00673 and autophagy are up-regulated in cisplatin-resistant NSCLC cells

Since it has been discovered, the biological function of Linc00673 has been well investigated, including its role in tumor metastasis (Lu et al, 2017; Guan et al, 2019), proliferation (Huang et al, 2017; Schmidt et al, 2019), and senescence (Roth et al, 2018). Recent study has identified Linc00673-V4 as the prevailing isoform in lung adenocarcinoma, but the function of other isoforms remains largely unknown. To the best of our knowledge, there is no report about the participation of Linc00673 isoforms in therapeutic chemoresistance. In that case, we started our research by investigating whether Linc00673 was involved in NSCLC chemoresistance and which isoform contributed the most in NSCLC chemoresistance. First, we found that Linc00673 was up-regulated in cisplatin-resistant A549 cells (A549/CDDP) (Fig 1A). We wondered whether this phenomenon was repeatable in WT NSCLC cells, so we exposed the WT NSCLC cell lines A549, H1975 (Fig 1B) and H596 (Fig S1A) under cisplatin treatment in a concentration gradient way. We found that the expression of Linc00673 increased in a dose-dependent manner after treatment of cisplatin. We also observed that A549/CDDP exhibited enhanced LC3B-II transformation and decreased p62 protein level (Fig 1C). In WT NSCLC cell lines (Figs 1D and

S1B), we found that the autophagy was induced in response to cisplatin treatment. To further validate this finding and monitor autophagic flux, stubRFP-sensGFP-LC3 vectors were transfected into cells. We observed significantly increased ratio of autophagolysosomes to autophagosomes in A549/CDDP (Figs 1E and S1C) and WT NSCLC cells under the treatment of cisplatin (Figs 1F and S1D–F).

So far, five different isoforms of Linc00673 have been reported in NSCLC, each may have distinctive biological function (Fig S1G) (Guan et al, 2019). We conducted further investigations to determine which isoform responded to cisplatin treatment and correlated with the heightened autophagy activity. The relative expression level of each isoform in WT A549 cells was in consistence with the previous study, as Linc00673-V3 to V5 had relatively high expression level (Guan et al, 2019) (Fig 1H). Intriguingly, Linc00673-V3 was significantly up-regulated in A549/CDDP cells compared with its parental cells, whereas the expression level of Linc00673-V4 and V5 remained unchanged (Fig 1G). Furthermore, similar phenomenon was also observed in WT A549 cells treated with cisplatin (Fig 1H). Considering the potential correlation between Linc00673 and autophagy, we treated WT A549 cells with rapamycin to induce autophagy and we found that Linc00673-V3 was significantly up-regulated by treatment of rapamycin (Fig 1I), which indicated that Linc00673-V3 could be induced upon autophagy induction and may be crucial for autophagy activation. Collectively, these data demonstrated that autophagy and Linc00673-V3 were up-regulated in response to cisplatin treatment in NSCLC cells.

### Linc00673-V3 regulated autophagy contributes to chemoresistance of NSCLC cells in vitro

Having found that Linc00673-V3 displayed the most striking change among the five reported isoforms, we next detected its expression in various NSCLC cell lines. We found that Linc00673-V3 was highly expressed in A549 and H1975 cells, whereas its expression level in H1299 and H1650 were relatively low. In consistence with the results above, A549/CDDP cells had the highest expression level of Linc00673-V3 (Fig 2A). Next, we transfected A549 and H1975 cells that presented relatively high expression levels of Linc00673-V3 with two individual siRNAs against Linc00673-V3 that were validated by our previous works (Lu et al, 2017; Wu et al, 2020) (Fig S2A). Conversely, Linc00673-V3 overexpression assay was performed in H1299 cells which had relatively low Linc0067-V3 expression level, and the fold-change was 148 post overexpression (Fig S2B). As shown in Fig 2B and C, Linc00673-V3 overexpression significantly enhanced the resistance of H1299 and H596 cells towards the cytotoxicity of cisplatin, whereas knockdown Linc00673-V3 in A549/CDDP, A549, and H1975 cells significantly reduced the resistance (Fig 2D–F). Considering the essential role of autophagy in chemoresistance, we treated NSCLC cells with autophagy inhibitor chloroquine (CQ) and 3-methyladenine (3-MA). When autophagy was blocked, NSCLC cells were more sensitive to the cytotoxicity of cisplatin (Fig 2G–I). Taken together, these data suggested that Linc00673-V3 associated autophagy activity participated in the chemoresistance of NSCLC cells.

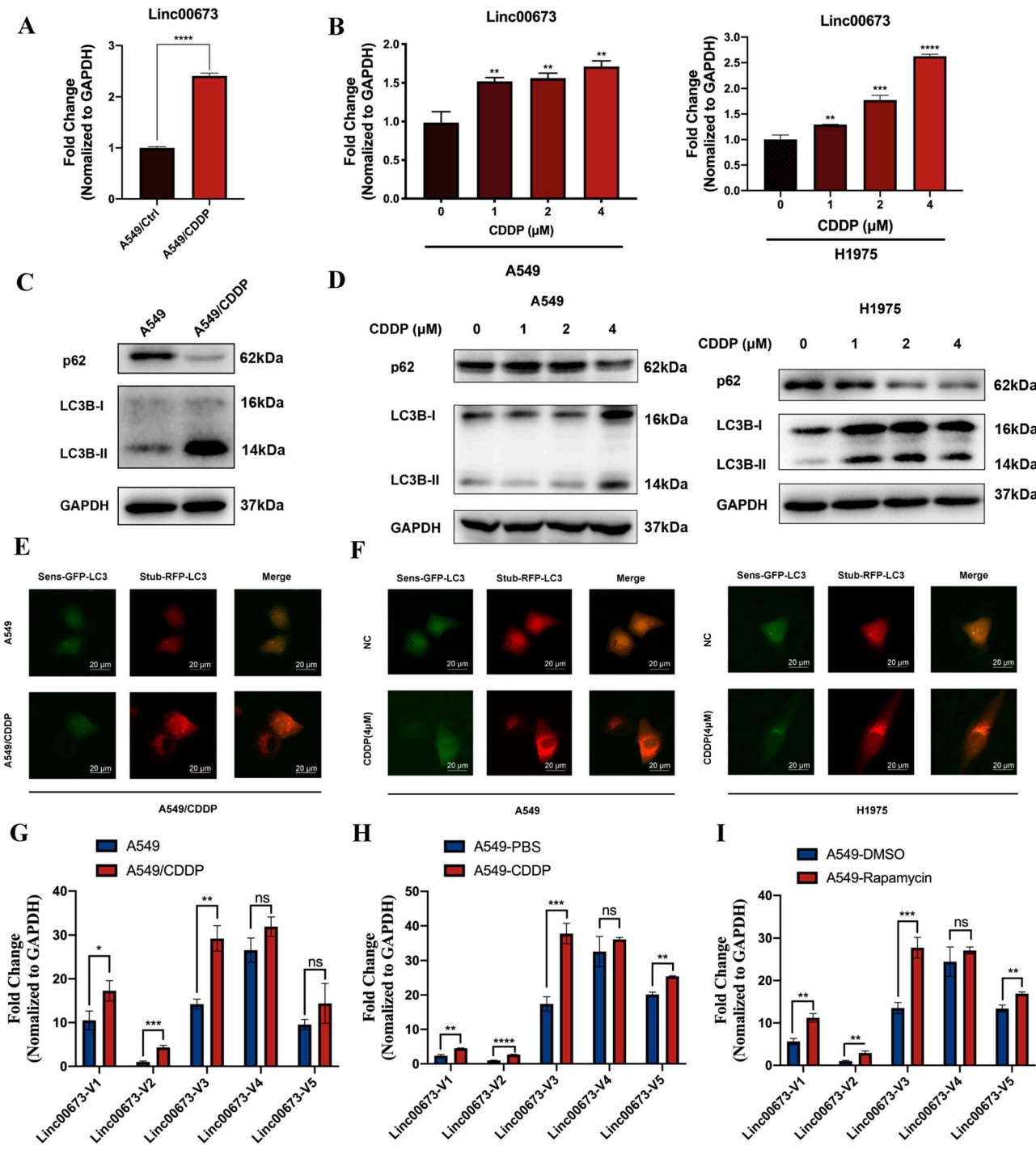

**Figure 1. Linc00673 and autophagy activity are elevated under cisplatin stimulation.**
**(A, B)** Linc00673 expression in A549/CDDP, parental A549 cells, and cisplatin-treated A549 or H1975 cells were detected by RT-qPCR. **(C)** LC3B and p62 protein were detected by Western blot analysis in A549/CDDP cells and parental A549 cells. **(D)** A549 or H1975 cells were treated with cisplatin for 24 h, LC3B and p62 protein were detected by Western blot analysis. **(E)** A549/CDDP cells or parental A549 cells were transfected with stubRFP-sensGFP-LC3 vectors, images were captured by fluorescence microscopy. Scale bar: 20 μm. **(F)** A549 or H1975 cells were transfected with stubRFP-sensGFP-LC3 vectors and treated with 4 μM cisplatin for 24 h. Images were captured by fluorescence microscopy. Scale bar: 20 μm. **(G)** Linc00673 isoforms in A549/CDDP cells and parental A549 cells were detected by RT-qPCR. **(H)** A549 cells were treated with or without 4 μM cisplatin for 24 h, Linc00673 isoforms were detected by RT-qPCR. **(I)** A549 cells were treated with or without 1 μM rapamycin for 24 h, Linc00673 isoforms were detected by RT-qPCR. Error bars indicate the mean ± SD, *$P < 0.05$, **$P < 0.01$, ***$P < 0.001$, ****$P < 0.0001$.

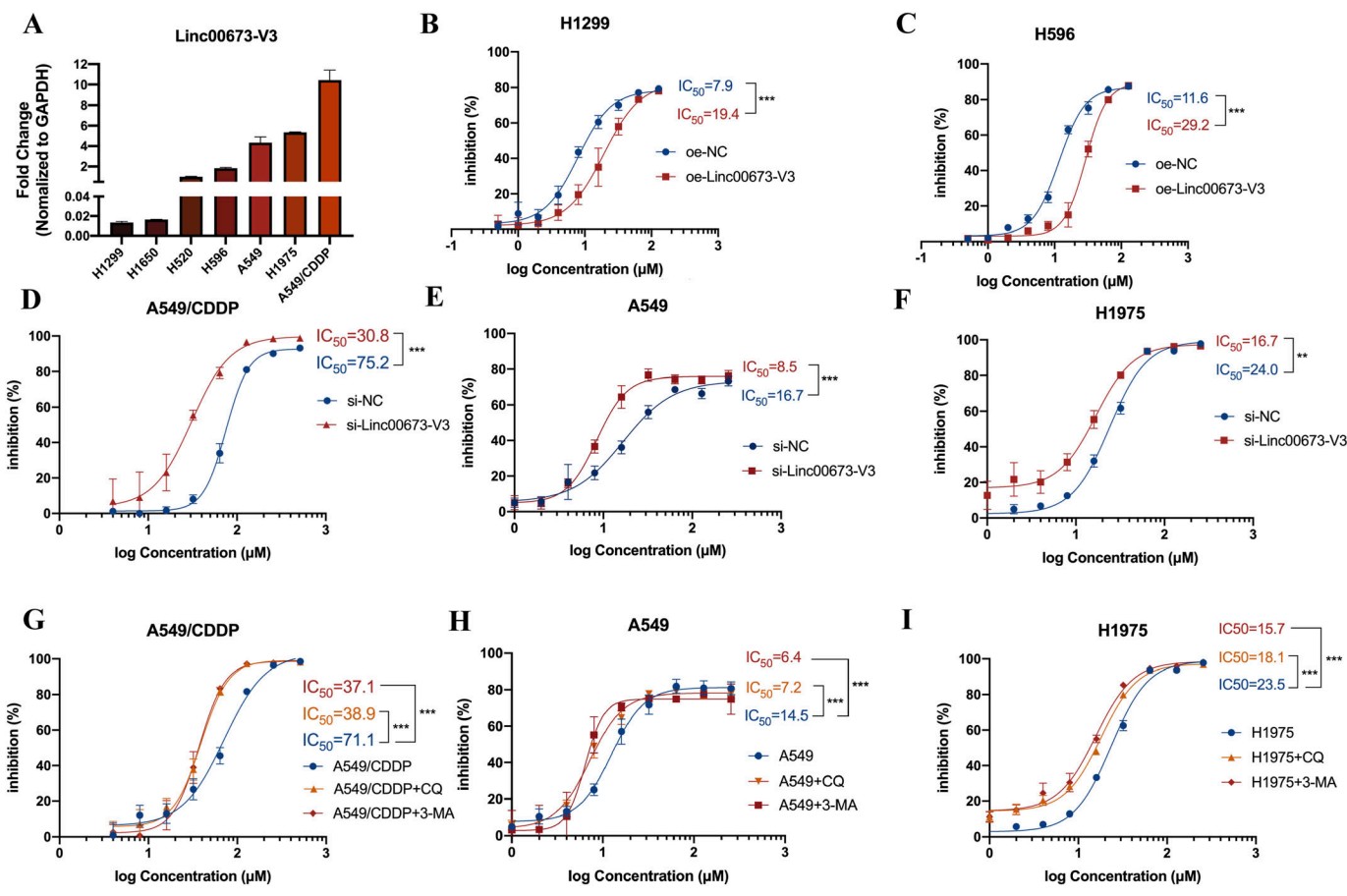

**Figure 2. Linc00673 and autophagy are required for chemoresistance.**
**(A)** Linc00673-V3 expression in seven NSCLC cell lines as determined by RT-qPCR. **(B, C)** The sensitivity of H1299 or H596 cells under Linc00673-V3 overexpression was determined by CCK-8 assay. **(D, E, F)** The sensitivity of A549/CDDP cells, A549, and H1975 cells under Linc00673-V3 knockdown was determined by CCK-8 assay. **(G, H, I)** The sensitivity of A549/CDDP cells, A549, and H1975 under CQ (5 $\mu$M, 24 h) or 3-MA (2.5 mM, 24 h) treatment were determined by CCK-8 assay. Error bars indicate the mean ± SD, *$P$ < 0.05, **$P$ < 0.01, ***$P$ < 0.001, ****$P$ < 0.0001.

## Linc00673-V3 increases the transcription of *LC3B*

To further investigate the role of Linc00673-V3 in autophagy, we knocked down Linc00673-V3 in A549 and H1975 cells and found the protein level of both LC3B-I and LC3B-II significantly decreased (Figs 3A and S3A). Similar phenomenon was validated by immunofluorescence analysis, which showed decreased LC3B puncta formation after knockdown of Linc00673-V3 (Figs 3B and S3B). Moreover, rapamycin was unable to increase LC3B-II level in neither A549 nor H1975 cells knocked down by Linc00673-V3 (Fig 3A and B). There were two possibilities for this phenomenon. One was that knockdown of Linc00673-V3 may accelerate autophagic degradation, which led to the reduced protein level of LC3B. Another possibility was that knockdown of Linc00673-V3 could directly inhibit the initiation of autophagic process. We first treated cells with CQ to study the autophagic flux. Cells treated with CQ alone exhibited massive LC3B-II accumulation, whereas cells treated with both si-Linc00673-V3 and CQ did not show the same phenomenon (Figs 3C and S3H and I), suggesting the impairment of autophagosome biogenesis upon Linc00673-V3 knockdown. Conversely, when cells were overexpressed with Linc00673-V3, the protein level of LC3B was strikingly increased

(Fig 3D). Moreover, LC3B lipidation and p62 degradation were markedly enhanced (Fig 3D) and immunofluorescence analysis suggested increased LC3B puncta formation after overexpressing Linc00673-V3 (Figs 3E and S3C). These findings together suggested that Linc00673-V3 was involved in autophagosome biogenesis and this may associate with the protein level of LC3B. To verify which isoform of Linc00673 was involved in autophagy, we knocked down overall Linc00673 and then overexpressed the five isoforms, respectively. We found that out of the five isoforms, only overexpression of Linc00673-V3 could significantly rescue the impairment of autophagy due to Linc00673 knockdown (Figs 3F and S3D).

Because it has been proved that knockdown of Linc00673-V3 could inhibit the initiation of autophagy, we further explored the potential mechanism underlying this phenomenon. We detected the expression of a series of ATG proteins required for autophagosome biogenesis and found that only LC3B was significantly reduced after knockdown of Linc00673-V3 (Figs 3G and S3J). We further detected the mRNA level of each gene and observed consistent result (Figs 3H and S3K). As we found the up-regulation of Linc00673-V3 under the treatment of cisplatin and the association between Linc00673-V3 and *LC3B* mRNA level (Fig S3E), we further explore the putative correlation

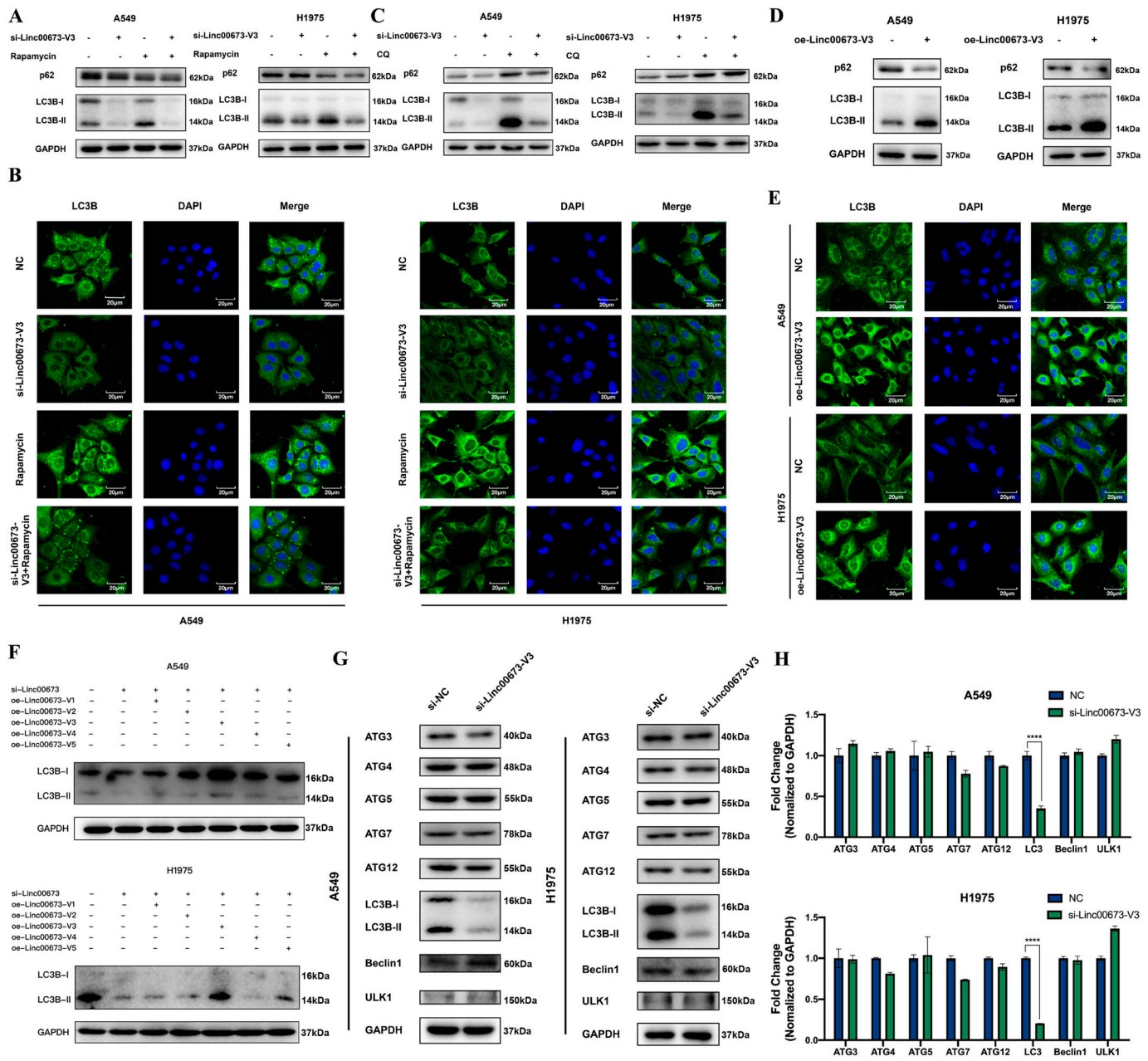

**Figure 3. Linc00673-V3 promotes NSCLC autophagy by enhancing *LC3B* transcription.**
**(A, C)** LC3B and p62 were detected in A549 and H1975 cells under the treatment of 1 μM rapamycin or 5 μM CQ for 24 h and Linc00673-V3 knockdown.
**(B)** Immunofluorescent staining was used to detect LC3B in A549 and H1975 cells under the treatment of 1 μM rapamycin for 24 h and Linc00673-V3 knockdown. Scale bar: 20 μm. **(D)** LC3B and p62 were detected in A549 and H1975 cells when Linc00673-V3 was overexpressed. **(E)** Immunofluorescent staining was used to detect LC3B in A549 and H1975 cells when Linc00673-V3 was overexpressed. Scale bar: 20 μm. **(F)** Total Linc00673 was knockdown and each isoform was overexpressed respectively to determine their effect on LC3B expression. **(G, H)** The protein level and mRNA level of autophagy-related genes under the treatment of Linc00673-V3 knockdown were determined by Western blot and RT-qPCR. Error bars indicate the mean ± SD, *P < 0.05, **P < 0.01, ***P < 0.001, ****P < 0.0001.
Source data are available for this figure.

between cisplatin treatment and *LC3B* transcription. The results showed that compared with the parental cells, A549/CDDP had higher *LC3B* transcript level (Fig S3F). When treated WT NSCLC cells with cisplatin, the mRNA level of *LC3B* were also elevated (Fig S3G). Taken together, these data suggested that Linc00673-V3 may participate in *LC3B* transcription regulation in NSCLC.

## Linc00673-V3 directly interacts with Smad3 in NSCLC

To elucidate the underlying mechanism of which Linc00673-V3 up-regulated the mRNA level of *LC3B* in NSCLC, we performed RNA affinity pulldown (RAP) assay to identify the proteins that directly interacted with Linc00673-V3. The RNA-binding proteins (RBP)

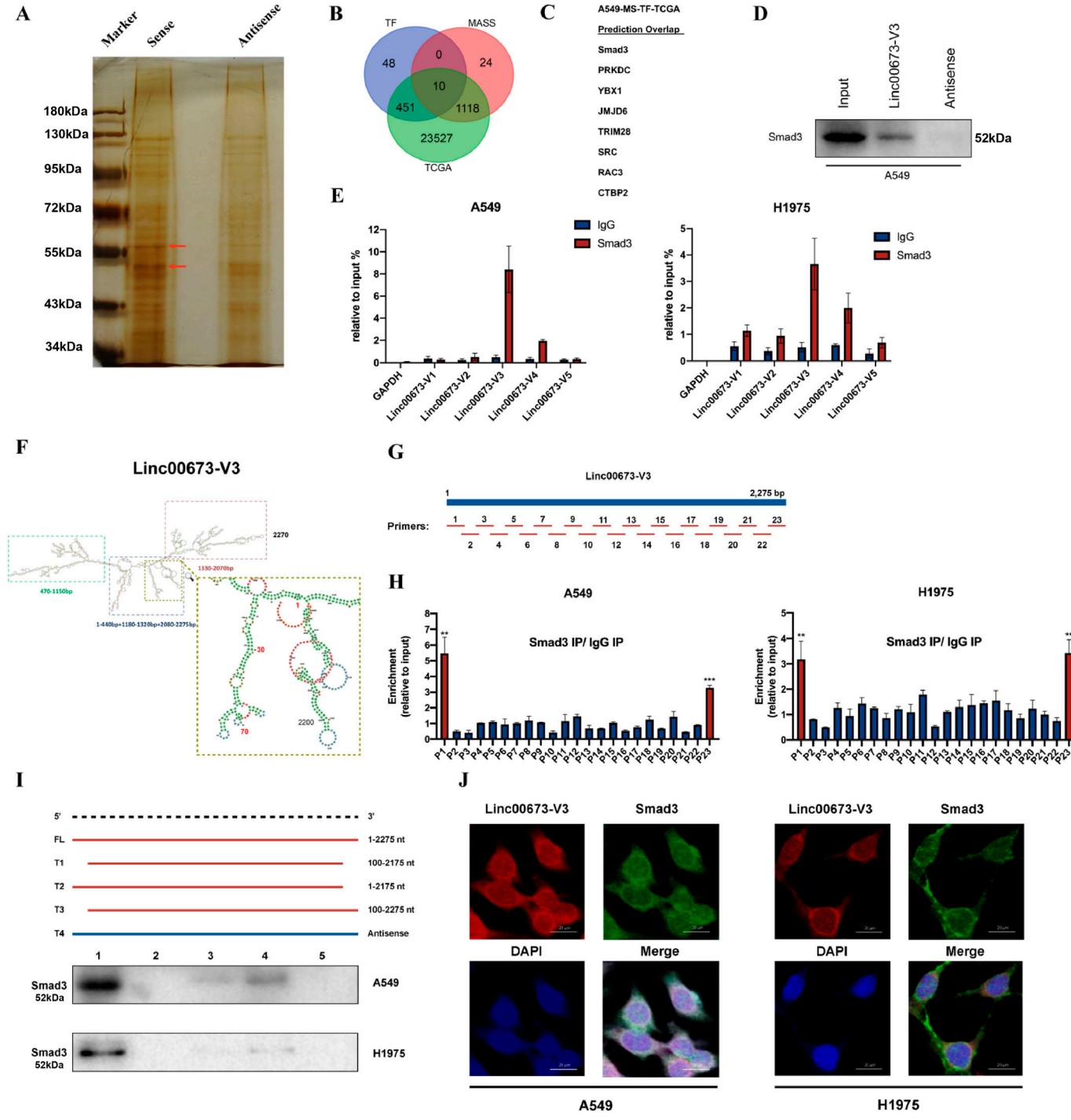

**Figure 4. Linc00673-V3 interacts with Smad3.**
**(A)** Result of RNA affinity pulldown followed by silver staining. **(B)** Venn diagram to demonstrate the overlapping of candidate genes. **(C)** The list of eight overlapped candidate genes. **(D)** Western blot validation of Smad3 pulled down by Linc00673-V3. **(E)** RNA immunoprecipitation assay verified the interaction between Smad3 and Linc00673-V3. **(F)** The predicted second structure of Linc00673-V3. **(G)** Schematic diagram of primers designed for CLIP-qPCR assay. **(H)** CLIP-qPCR assay to determine the direct binding fragments of Linc00673-V3 with Smad3. **(I)** Truncation mutations of Linc00673-V3 and RNA affinity pulldown results. **(J)** FISH assay to determine the colocolization of Linc00673-V3 and Smad3. Scale bar: 20 μm. Error bars indicate the mean ± SD, *P < 0.05, **P < 0.01, ***P < 0.001, ****P < 0.0001.

pulled down by biotin-labeled sense- or antisense-Linc00673-V3 were analyzed by silver staining and the bands around 55KD were different (as indicated by red arrows) (Fig 4A). The pulldown RBPs were then analyzed by mass spectrometry (Table S1). We overlapped the pulldown proteins with genes that were highly expressed in NSCLC tissues from TCGA and potential upstream

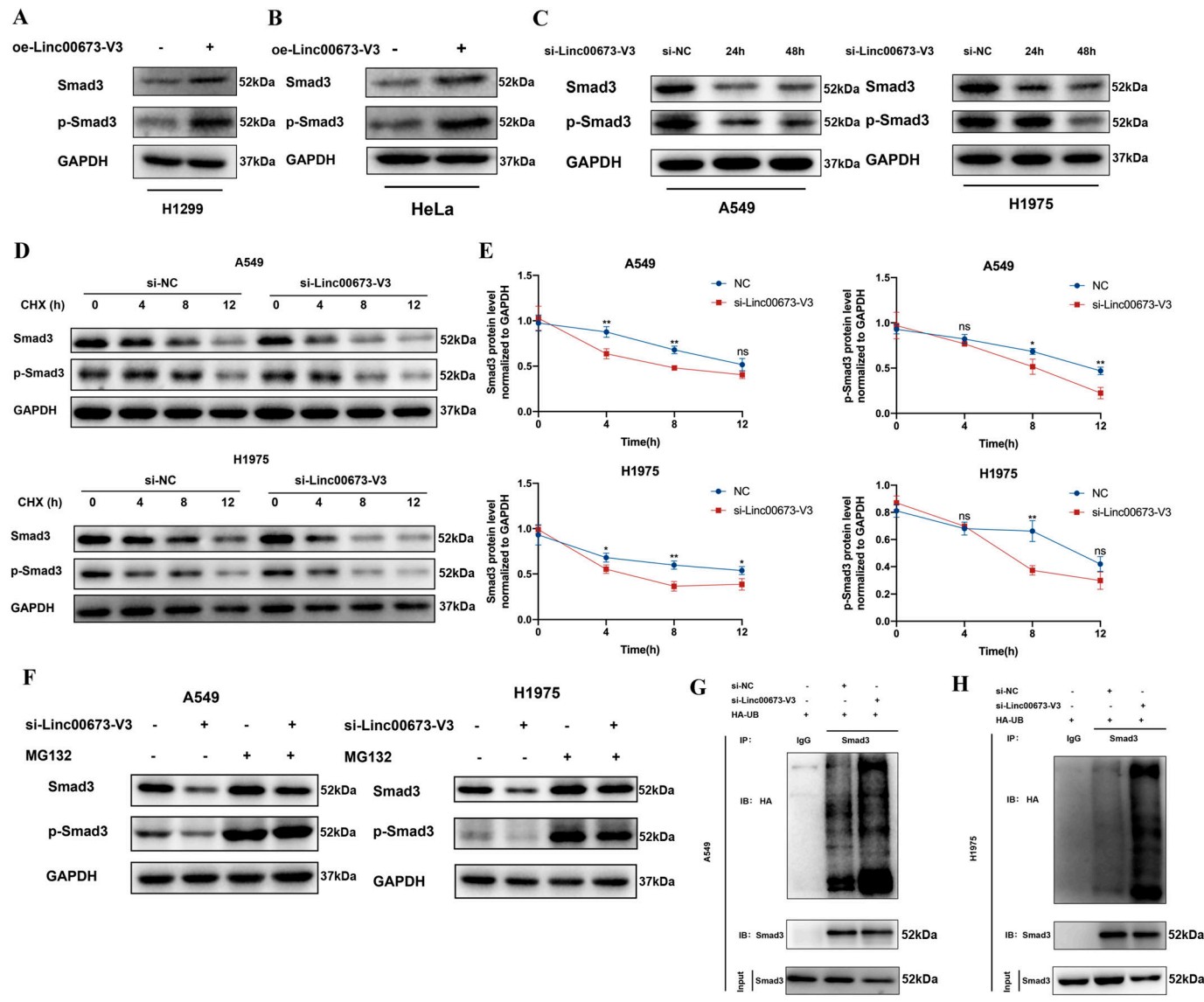

**Figure 5. Linc00673-V3 prevents Smad3 from ubiquitin-proteasome degradation.**
**(A, B)** Detection of Smad3 and p-Smad3 expression after Linc00673-V3 overexpression by Western blot. **(C)** Detection of Smad3 and p-Smad3 expression after Linc00673-V3 knockdown by Western blot. **(D)** Detection of Smad3 and p-Smad3 degradation rates under the treatment of 50 µg/ml cycloheximide for the indicated time after Linc00673-V3 knockdown. **(E)** Quantification of Smad3 and p-Smad3 degradation rates. **(F)** Detection of Smad3 and p-Smad3 degradation under the treatment of 20 µM MG132 for 12 h after Linc00673-V3 knockdown. **(G, H)** Smad3 ubiquitination was detected after Linc00673-V3 knockdown.

transcriptional factors of *LC3B* predicted by AnimalTFDB, a comprehensive transcription factor database (Fig 4B) (Hu et al, 2019). It turned out that there were eight candidate proteins that met the above screening criteria (Fig 4C). We were particularly interested in Smad3 because its well-known function as a transcription factor and its implications in autophagy (Shim et al, 2021; Yang et al, 2021), whereas its exact functions and underlying mechanisms in autophagy remain unclear.

We subsequently tested whether Smad3 was participated in Linc00673-V3 induced autophagy and chemoresistance in NSCLC. First, we validated the interaction between Linc00673-V3 and Smad3 in all the RBPs. It turned out that there was a significant enrichment in the samples pulled down by the Linc00673-V3 sense sequence but not the antisense sequence (Fig 4D). Consistently, the results of RNA immunoprecipitation (RIP) assay demonstrated greater enrichment of Linc00673-V3 captured by Smad3 specific antibody (Fig 4E). It was also noteworthy that there was slight enrichment of other isoforms. Given the similarity of the sequence and structure of those isoforms, such enrichment was also reasonable.

Based on the minimum free energy fold algorithm, we used the bioinformatic tool RNAfold to predict the secondary structure of Linc00673-V3 (Fig 4F). To identify the exact fragment that bound to Smad3, cross-linking RIP followed by RT-qPCR (CLIP-qPCR) was performed. As shown in Fig 4G, 23 RT-qPCR primers were designed for every 100-nt region of Linc00673-V3 (primers used in CLIP assay

were listed in Table S2) and the CLIP-qPCR results showed that primer pair 1 and pair 23 were successfully amplified in both A549 and H1975 cells (Fig 4H), which were referred to as the first 1–100 and last 2,200–2,275 fragments accordingly. These two amplified regions were shown in the secondary structure of Linc00673-V3 (Fig 4F), and we found that these two regions were spatially close and formed stem-loop structures, which may partially explain the results of CLIP-qPCR assay. To further validate the exact region of Linc00673-V3 bound to Smad3, vectors carrying full-length or truncation mutations of Linc00673-V3 (V3-FL [1–2,275 nt], V3-T1 [100–2,175 nt], V3-T2 [1–2,175 nt], V3-T3 [100–2,275 nt], and V3-T4 [antisense sequence]) were constructed (Fig 4I). RAP assay was performed, only full-length Linc00673-V3 could bind to Smad3 (Fig 4I), suggesting that both 1–100-nt and 2,175–2,275-nt regions were essential for the interaction of Linc00673-V3 with Smad3.

Subcellular localization of LncRNA was thought to be correlated with its biological functions, thus RNA FISH and cellular fractionation assay were performed. Consistent with previous studies, Linc00673-V3 was mostly found in the cytoplasm of NSCLC cells (Fig S4A) and the results of RNA FISH also indicated the potential colocalization of Linc00673-V3 and Smad3 in cytoplasm (Fig 4J). Considering the transcriptional regulation capacity of Smad3, we further investigated whether Linc00673-V3 and Smad3 would regulate each other's transcription reciprocally. We found that there was no significant difference in transcriptional level of Linc00673-V3 after Smad3 knockdown or overexpression (Fig S4B). Besides, Linc00673-V3 knockdown or overexpression also did not alter the transcriptional level of Smad3 (Fig S4C). Collectively, these results indicated the direct binding of Linc00673-V3 to Smad3 protein, and this interaction might be mediated by the stem-loop regions formed by the first 1–100-nt and the last 2,200–2,275-nt fragments of Linc00673-V3.

### Linc00673-V3 elevates Smad3 protein level via inhibiting its ubiquitination in NSCLC

As Linc00673-V3 and Smad3 did not influence each other's transcription, we further explore whether Linc00673-V3 could influence the Smad3 protein level. We observed the increased protein level of Smad3 after overexpressing Linc00673-V3 in H1299 cells (Fig 5A). Given that phospho-Smad3 (p-Smad3, Ser423/425) is its active form in terms of transcriptional regulation, we further detected the protein level of p-Smad3 in H1299. It turned out that overexpressing Linc00673-V3 could also increase the protein level of p-Smad3 (Fig 5A). Similar result was observed in HeLa cells (Fig 5B), which suggested that Linc00673-V3 could also influence the protein level of Smad3 in other cancer types. Consistently, Linc00673-V3 knockdown led to reduced protein levels of both Smad3 and p-Smad3 in A549 and H1975 cells (Fig 5C). These data thus indicated a positive regulatory role of Linc00673 in Smad3 protein level.

Because Linc00673-V3 has no effects on the transcription of Smad3 (Fig S4C), we thus asked whether it influenced the degradation of Smad3. To this end, we treated A549 and H1975 cells with cycloheximide to block de novo protein synthesis and found that knockdown of Linc00673-V3 significantly accelerated the degradation of both Smad3 and p-Smad3 (Fig 5D and E). Ubiquitin–proteasome system and autophagy are two major protein quality

control pathways responsible for protein degradation and intracellular homeostasis (Silva-Fisher et al, 2020; Lu et al, 2021). We treated cells with the autophagy inhibitor CQ or the proteasome inhibitor MG132. Treatment of CQ did not attenuate the degradation of either Smad3 or p-Smad3 caused by Linc00673-V3 knockdown (Fig S5A and B). However, treatment of MG132 strikingly restored the protein levels of both Smad3 and p-Smad3 (Fig 5F), indicating Linc00673-V3 may stabilize Smad3 via preventing its degradation by the ubiquitin–proteasome system. As expected, the ubiquitination level of Smad3 dramatically increased after Linc00673 knockdown (Figs 5G and H and S5C and D). Taken together, these data demonstrated that the interaction between Linc00673-V3 and Smad3 could inhibit the ubiquitination of Smad3 and prevent Smad3 degradation through the ubiquitin–proteasome system.

### Linc00673-V3 stabilizes Smad3 by counteracting the interaction between Smad3 and STUB1

Having established that Linc00673-V3 modulated Smad3 ubiquitination and degradation, we next aimed to study whether Linc00673-V3 disrupted the activity of certain E3 ligase that targeted Smad3. Several E3 ligases including Nedd4L (Gao et al, 2009; Louzada et al, 2021), STUB1 (Li et al, 2004; Shang et al, 2014; Shi et al, 2020), and VHL (Wang et al, 2016; Zhou et al, 2022) have been reported to target Smad3 for ubiquitination. To verify their effects on Smad3 protein level, siRNA has been designed to target each E3 ligase (Fig S6A and B). Knockdown of STUB1 increased Smad3 in A549 and H1975 cells (Fig 6A and B), whereas knockdown of Nedd4L or VHL had marginal impact on Smad3 protein level in A549 and H1975 cells (Fig S6C–F). Importantly, knockdown of STUB1 strongly increased the protein level of both LC3B-I and LC3B-II (Fig 6A and B). Moreover, the transcription level of *LC3B* also increased after STUB1 knockdown in A549 and H1975 cells (Fig 6C). Because Linc00673-V3 inhibited the ubiquitination of Smad3, we thus asked whether it disrupted the interaction of Smad3 with STUB1. To this end, we performed immunoprecipitation assay and confirmed the interaction between Smad3 and Nedd4L, STUB1, or VHL (Fig 6D and E). However, the interaction between Smad3 and STUB1 was markedly enhanced upon Linc00673-V3 knockdown in A549 and H1975 cells, whereas the interaction of Smad3 with the other two E3 ligases remained the same under the same conditions (Fig 6D and E). Furthermore, overexpression of STUB1 significantly increased the ubiquitination level of Smad3, whereas Linc00673-V3 overexpression attenuated the effect of STUB1 (Fig 6F and G). Taken together, these results suggested that Linc00673-V3 and STUB1 competitively bound to Smad3, and Linc00673-V3 stabilized Smad3 by preventing STUB1-mediated ubiquitination and proteasome degradation of Smad3.

### *LC3B* is transcriptionally regulated by Smad3 and knockdown of Smad3 suppresses autophagy-associated chemoresistance in NSCLC

Using the bioinformatic tool JASPAR (Fornes et al, 2020), we found two putative Smad3-binding sites in the upstream promoter region of *LC3B* and they were referred to as region (a) and region (b) as shown in Fig 7A. We then cloned the *LC3B* promoter region and

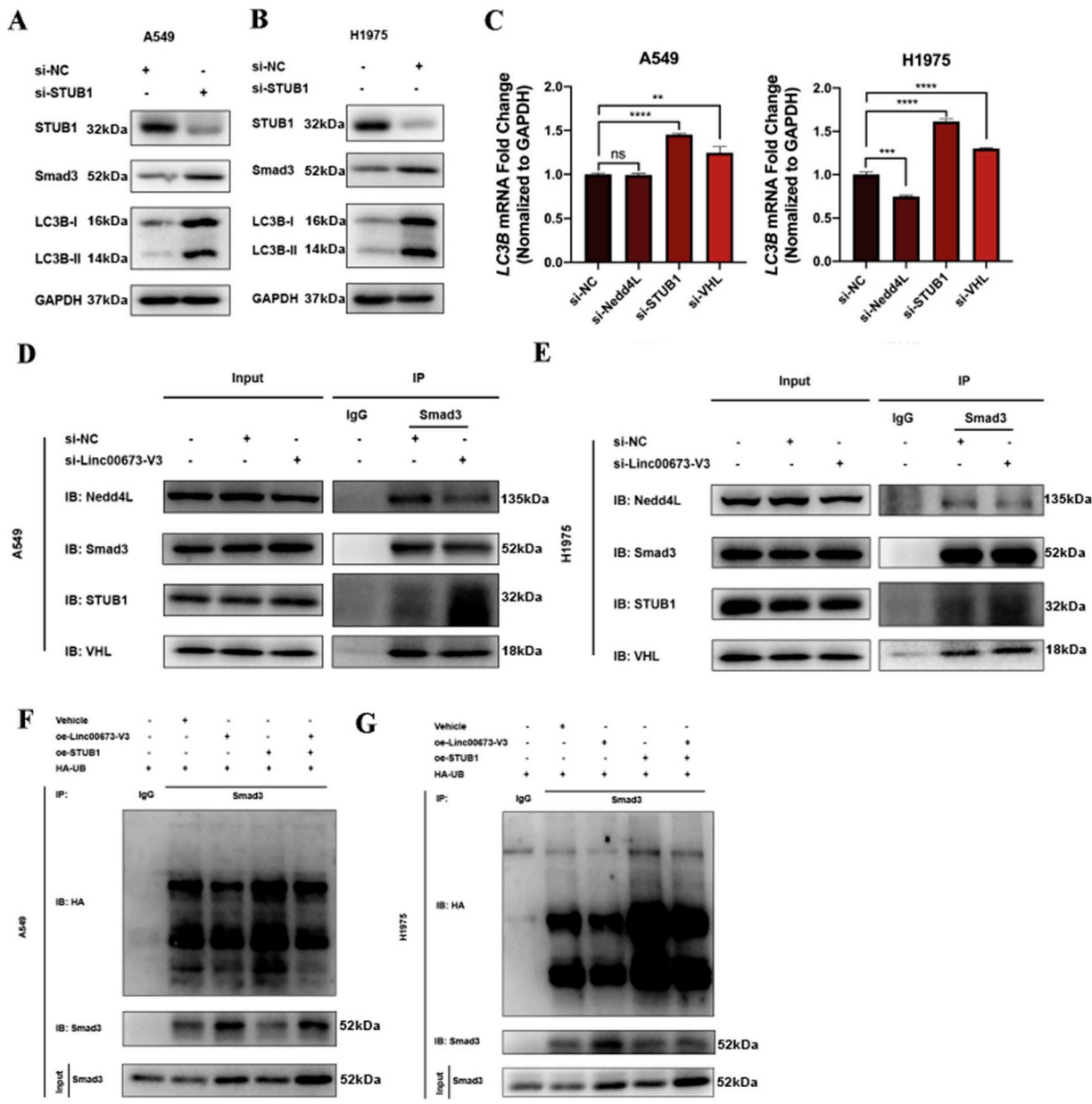

**Figure 6. Linc00673-V3 prevents Smad3 from STUB1-mediated ubiquitination.**
**(A, B)** Detection of Smad3 and LC3B after STUB1 knockdown in A549 and H1975 cells. **(C)** Detection of *LC3B* transcription level after STUB1 knockdown in A549 and H1975 cells. **(D, E)** Interaction between Smad3 and Nedd4L, STUB1, or VHL after Linc00673-V3 knockdown in A549 and H1975 cells was detected by immunoprecipitation. **(F, G)** Smad3 ubiquitination was detected after STUB1 or Linc00673-V3 overexpression in A549 or H1975 cells. Error bars indicate the mean ± SD, *$P$ < 0.05, **$P$ < 0.01, ***$P$ < 0.001, ****$P$ < 0.0001.

constructed the luciferase reporter plasmid pGL4-*LC3B* containing region (a) and region (b) (Fig 7A). The results of dual-luciferase reporter assay showed that overexpression of Smad3 significantly increased the luciferase activity, and this effect was much more striking when a constitutively activated form of Smad3 (Smad3-2SD) was overexpressed (Fig 7B). Consistently, the luciferase activity induced by Smad3 overexpression was strongly suppressed upon treatment of SIS3, a selective inhibitor of Smad3 (Fig 7B).

To figure out the exact binding site of p-Smad3 in the promoter region of *LC3B*, chromatin immunoprecipitation (ChIP-qPCR) assay

was performed. In the absence of additional stimulations, we found the binding of p-Smad3 to region (a) or (b) had no significant difference (Figs 7C and S7A). However, under the treatment of cisplatin (Figs 7D and S7B) or rapamycin (Figs 7E and S7C), the region (b) showed a greater binding capacity with p-Smad3 than region (a). In addition, a reference region (c) within the promoter region of *LC3B* was chosen to verify the specificity of the interaction of p-Smad3 with region (b). As shown in Fig S7D and E, there was no significant binding in region (c). Based on these findings, we then cloned the (b) region and constructed a luciferase reporter plasmid,

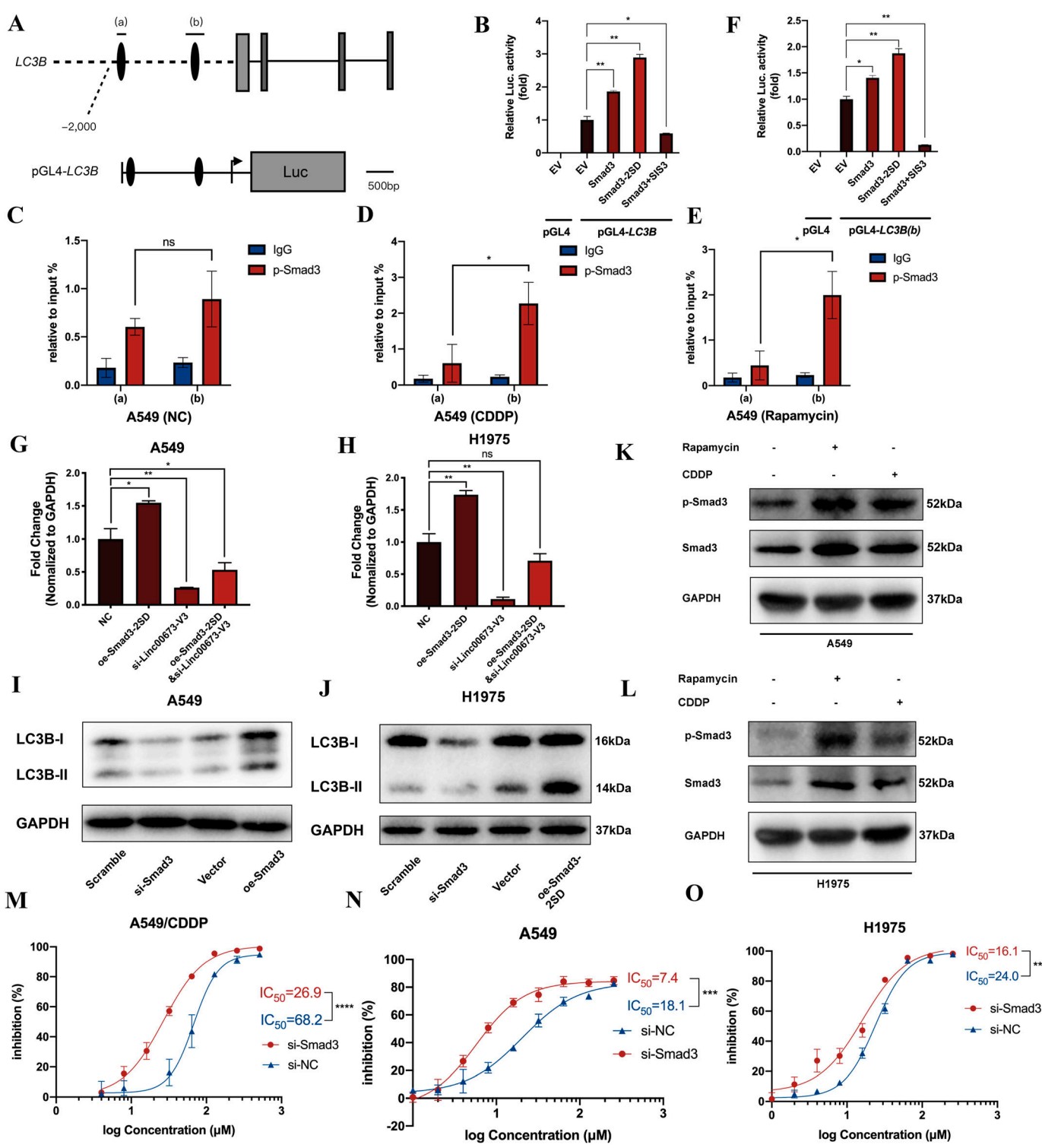

**Figure 7. Smad3 directly promotes *LC3B* transcription.**
**(A)** Schematic diagram of the promoter region of *LC3B* gene and the structure of luciferase reporter vector. **(B, F)** WT Smad3, Smad3-2SD, or Smad3 inhibitor SIS3 were applied with pGL4-*LC3B* or pGL4-*LC3B(b)* reporter vector in H293T cells. Luciferase activity was detected 24 h after transfection using the dual-luciferase assay. **(C, D, E)** Interaction between p-Smad3 and promotor regions of *LC3B* was detected using CHIP-qPCR assay under the treatment of cisplatin or rapamycin in A549 cells. **(G, H)** *LC3B* transcription level was detected in A549 or H1975 cells under the treatment of Smad3-2SD overexpression or Linc00673-V3 knockdown. **(I, J)** LC3B protein was detected in A549 and H1975 cells under the treatment of Smad3-2SD overexpression or Linc00673-V3 knockdown. **(K, L)** Protein level of p-Smad3 was detected under the treatment of 4 μM cisplatin or 1 μM rapamycin for 24 h. **(M, N, O)** The sensitivity of A549/CDDP cells, A549 and H1975 cells under Smad3 knockdown was determined by CCK-8 assay. Error bars indicate the mean ± SD, *P < 0.05, **P < 0.01, ***P < 0.001, ****P < 0.0001.

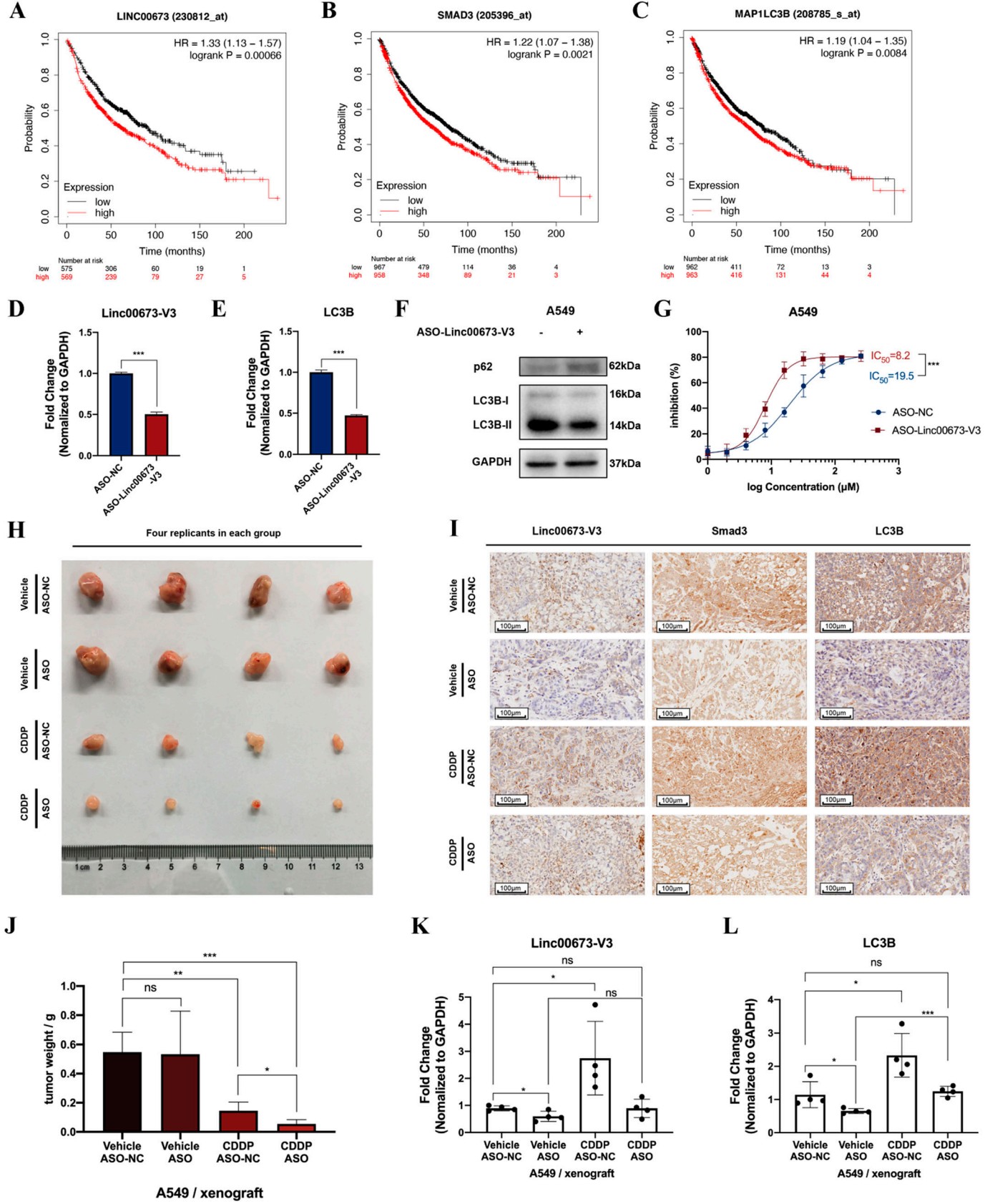

pGL4-*LC3B(b)*, and the results of dual-luciferase assay also confirmed the interaction between region (b) and Smad3 (Fig 7F).

Apart from dual-luciferase assay and ChIP-qPCR, we directly detected the *LC3B* transcription level in A549 and H1975 cells. After overexpressing Smad3-2SD, the transcriptional level of *LC3B* was elevated, and it could partially attenuate the down-regulation of *LC3B* mRNA level caused by Linc00673-V3 knockdown (Fig 7G and H). The protein level of LC3B was also detected, and we found that knockdown of Smad3 led to down-regulation of both LC3B-I and LC3B-II protein level, whereas overexpression of Smad3-2SD caused the accumulation of LC3B-I, which were in consistence with the up-regulation of *LC3B* transcriptional level (Fig 7I and J). We then detected the protein level of p-Smad3, and it appeared that p-Smad3 increased under the treatment of cisplatin or rapamycin (Fig 7K and L). These results suggested that Smad3-mediated transcription of *LC3B* was dependent on Linc00673-V3 upon autophagy induction.

Because we had established that Smad3 could regulate *LC3B* gene expression and thus affected autophagy, we further explored its role in NSCLC chemoresistance. The results showed that knockdown of Smad3 increased the sensitivity of NSCLC cells to cisplatin (Fig 7M–O). Taken together, these data suggested that Linc00673-V3–mediated Smad3 activation could directly up-regulate *LC3B* gene expression and participated in autophagy-associated chemoresistance of NSCLC.

### Linc00673-V3 correlates with poor prognosis of NSCLC patients and Linc00673-V3–targeted antisense oligonucleotide suppresses autophagy-associated chemoresistance in vivo

Data from KM plotter demonstrated that higher Linc00673, Smad3, and MAP1LC3B (LC3B) expression levels were correlated with worse overall survival outcomes (Fig 8A–C), and prognosis data from GSE37745 also validated this finding (Fig S8A and B). To investigate the effects of Linc00673-V3 on NSCLC chemoresistance in vivo, tumor xenograft nude mouse model has been established. Moreover, specific antisense oligonucleotide (ASO) of Linc00673-V3 (ASO-Linc00673-V3) has been designed to explore the potential role of Linc00673-V3 as a therapeutic target in NSCLC. In vitro validation confirmed the efficiency of ASO as it decreased Linc00673-V3 and *LC3B* expression level (Fig 8D and E). LC3B lipidation was also decreased after treatment of ASO-Linc00673-V3, which was accompanied by accumulation of p62 (Fig 8F). As expected, treatment with ASO-Linc00673-V3 enhanced the sensitivity of A549 cells towards cisplatin in vitro (Fig 8G). In vivo validation demonstrated that both tumor volume and weight were significantly reduced in cisplatin-treated mice (Fig 8H and J). When combined cisplatin with ASO-Linc00673-V3, the tumor volume and weight further reduced compared with cisplatin treatment alone (Fig 8H and J), which

indicated that Linc00673-V3 knockdown increased the sensitivity of tumor cell towards cisplatin in vivo. IHC staining revealed enhanced Linc00673-V3, Smad3, and LC3B expression in tumor tissues when treated with cisplatin, which is largely attenuated by ASO-Linc00673-V3 (Fig 8I). RT-qPCR of tumor tissues further confirmed the reduced expression of Linc00673-V3 and *LC3B* when treated with ASO-Linc00673-V3 (Fig 8K and L). Taken together, these results validated the role of Linc00673-V3 in chemoresistance in vivo via up-regulation of autophagy and suggested that ASO-Linc00673-V3 could be a promising antitumor therapeutic strategy especially in patients with chemoresistance of cisplatin.

## Discussion

Chemoresistance occurs as innated or acquired feature of cancer cells during treatment and always causes therapeutic failure and tumor recurrence in patients with advanced NSCLC. Research has been focusing on the molecular mechanism behind this phenomenon and activated autophagy has been found to play a pivotal role in acquired chemoresistance (Cai et al, 2019; Ghafouri-Fard et al, 2022). Here, we report an oncogenic role of Linc00673-V3 in NSCLC chemoresistance as summarized in a schematic model in the Graphical Abstract. Upon cisplatin treatment, Linc00673-V3 is up-regulated and subsequently completes with STUB1 to bind Smad3, leading to stabilization of Smad3. Smad3 then functions as a transcription factor to drive the transcription of *LC3B*, resulting in activation of autophagy and ultimately chemoresistance in NSCLC. Importantly, ASO-Linc00673-V3 is able to suppress the expression of Linc00673-V3 and eventually suppress autophagy and chemoresistance. These findings together indicate Linc00673-V3 as a novel target to overcome chemoresistance in NSCLC.

Since its discovery, dysregulation of Linc00673 is closely related to tumorigenesis and malignant progression in several solid tumors (Huang et al, 2017; Lu et al, 2017; Guan et al, 2019). It is reported that lncRNA could participate in regulation of autophagy by multiple ways, with one of the most studied being lncRNAs acting as competitive endogenous RNAs (ceRNAs). They can bind to cytoplasmic miRNAs, thereby reducing the degradation of ATG gene mRNA (YiRen et al, 2017; Zhou et al, 2020; Luo et al, 2021). In addition to acting as ceRNAs, lncRNAs can also epigenetically regulate the expression of ATG genes. Wang et al (2019) reported that in ER-positive breast cancer cells, long non coding RNA H19 could inhibit DNA methyltransferase DNMT3B mediated methylation of key autophagy gene *Beclin1*'s promoter. Overexpression of H19 was observed in tamoxifen-resistant breast cancer cells and correlated with enhanced autophagy activity (Wang et al, 2019). What's more, Li et al (2004) reported that lncRNA NEAT1 functioned as an autophagy inducer by modulating the expression of multiple autophagy-

---

**Figure 8. Linc00673-V3 is associated with poor survival in NSCLC patients and is associated with chemoresistance in vivo.**
**(A, B, C)** Association of Linc00673, Smad3, and MAP1LC3B (*LC3B*) expression with overall survival of NSCLC patients. **(D)** Validation of antisense oligonucleotide (ASO) knockdown efficiency in A549 cells. **(E, F)** Detection of *LC3B* transcription level and LC3B protein after ASO transfection in A549 cells. **(G)** The sensitivity of A549 cells under ASO transfection was determined by CCK-8 assay. **(H)** The xenograft tumors in nude mice under different treatment were shown. **(I)** The Linc00673-V3 expression was determined by in situ hybridization, the protein level of Smad3 and LC3B were determined by immunohistochemical staining. Scale bar = 100 μm. **(J)** Tumor weights were measured. **(K, L)** Linc00673-V3 and *LC3B* expression in xenograft tumor tissues under different treatment was detected by RT-qPCR. Error bars indicate the mean ± SD, *P < 0.05, **P < 0.01, ***P < 0.001, ****P < 0.0001.

related genes including *ATG3*, *ATG5*, and *Beclin1*. Mechanistically, NEAT1 can alter the histone modification near the transcriptional start sites of these ATG genes and influence the recruitment of signal transducer and transcription factors to these gene promoters (Wang et al, 2022). However, to our knowledge, there are currently no reports regarding the involvement of Linc00673 in autophagy regulation. With in-depth analyses, Linc00673 was shown to have five different isoforms (as shown in Fig S1G). Guan et al (2019) reported that Linc00673-V4 was the most abundant transcript of Linc00673 in lung adenocarcinoma, which was further confirmed by our results (Fig 1G–I). The difference in sequences and secondary or higher structures of Linc00673 may explain the different functions of each isoform. In that case, the relative abundance of Linc00673 isoforms could be highly tissue-specific, but the molecular mechanisms underlying Linc00673 alternative splicing remains to be explored. Here, we reported that upon autophagy activation, the transcript level of Linc00673-V3 was significantly elevated. By restoration of different isoforms of Linc00673 in cells depleted of Linc00673, we showed that Linc00673-V3 played the major role in autophagy activation (Fig 3F), which favored cancer cell survival upon cisplatin treatment (Fig 2D–F).

Smad3 is known for its transcriptional regulation function in canonical TGF-$\beta$ signaling pathway. Apart from the canonical TGF-$\beta$ signaling pathway, recent study indicated that Smad2/3 could directly participate in primary cilia–mediated signaling transduction in trabecular meshwork cells, which influenced the LC3 protein level, but the exact mechanism remained to be elucidated (Shim et al, 2021). Up to date, the relationship between Smad family proteins and transcriptional regulation of ATG genes in lung cancer is still largely unknown. In this study, we discovered the capacity of Smad3 in transcriptional regulation of *LC3B* gene by a series of dual-luciferase reporter gene assays and ChIP assays (Fig 7B–F). We further showed that Linc00673-V3 stabilized Smad3 via competing with STUB1 to bind Smad3 and hence prevented its ubiquitination and the eventual proteasomal degradation (Fig 6D–G). Moreover, the binding of Linc00673-V3 to Smad3 and resultant transcription of *LC3B* could be directly induced by cisplatin or rapamycin (Figs 7D and E and S7B and C), suggesting a novel function of Smad3 in regulation of autophagy via a mechanism independent of the canonical TGF-$\beta$ signaling pathway.

In summary, we reported Linc00673-V3 isoform as a positive regulator of autophagy via Smad3-mediated *LC3B* transcription and the ultimate activation of autophagy, which contributed to chemoresistance in NSCLC. Targeting Linc00673-V3 is thus a novel strategy to overcome chemoresistance in NSCLC in the future.

# Materials and Methods

## Cell culture and reagents

A549, H1975, and cisplatin-resistant A549 (A549/CDDP) cell lines were cultured in RPMI-1640 medium (KEYI), supplemented with 10% FBS (SERRANA), 1% GlutaMAX (Gibco), and 1% sodium pyrurate (Gibco), final concentration of 100 U/ml penicillin (Gibco) and 100

U/ml streptomycin (Gibco). The A549/CDDP cells were purchased from the Cell Resource Center, Peking Union Medical College (PCRC), China. During culturing, the A549/CDDP cells were continuously exposed to 20 $\mu$M concentration of CDDP to maintain their resistance to cisplatin. H1299, H596, H1650, H520, H293T, and HeLa cell lines were cultured in DMEM medium (Wisent), supplemented with 10% FBS and final concentration of 100 U/ml penicillin (Gibco) and 100 U/ml streptomycin (Gibco). All cell lines were cultured at 37°C in an atmosphere containing 5% $CO_2$.

Reagents used in this study were as follows: autophagy inhibitors chloroquine (CQ) and 3-methyladenine (3-MA) (Selleck), autophagy activator rapamycin (Selleck), protein synthesis inhibitor cycloheximide, proteasome inhibitor MG132 (Selleck), and Smad3-specific inhibitor SIS3 (Selleck).

## Cell transfection

The siRNAs of Linc00673 and Smad3, purchased from GenePharma, were transfected into cells using PowerFect siRNA Transfection Reagent (SignaGen Laboratories). The nucleotide sequences of si-Linc00673-#3, si-Linc00673-#5, si-Smad3, si-Nedd4L, si-STUB1, and si-VHL were listed in Table S3. The sequences of Linc00673-V1, Linc00673-V2, Linc00673-V3, Linc00673-V4, Linc00673-V5, Smad3, and STUB1 were cloned into pcDNA3.1 vectors. The stubRFP-sensGFP-LC3 plasmid was kindly given by Jianbing Zhang from Zhejiang Provincial People's Hospital. In general, the stubRFP-sensGFP-LC3 plasmid codes the fusion protein components include the red fluorescent protein stubRFP, the GFP sensGFP, and the autophagy marker protein LC3. SensGFP is an acid-sensitive protein, whereas stubRFP is a stable fluorescent protein unaffected by pH changes. When the autophagosomes fuse with lysosomes to form autolysosomes, the sensGFP fluorescence is quenched and only red fluorescence punctate aggregates can be detected. The ratio of sensGFP to stubRFP bright spots was used to evaluate the progression of autophagic flux. All the plasmids were transfected with Lipofectamine 3000 (Thermo Fisher Scientific).

## Western blot analysis

Cells were lysed to extract proteins with lysis buffer (NCM biotech) containing protease and phosphatase inhibitors cocktail (NCM biotech). The proteins were separated by 8–13.5% SDS–PAGE gels and transferred to nitrocellulose filter membranes, and these membranes were incubated with primary antibodies overnight at 4°C, followed by incubation with anti-mouse or anti-rabbit HRP-conjugated secondary antibodies (1:5,000; Promega). Finally, the protein–antibody complex was detected with the ECL substrate (Cyanagen) by ChemiScope 3300 Mini (Clinx). Primary antibodies were as follows (detail information listed in Table S4): GAPDH (1: 5,000; Proteintech), LC3B (1:1,000; CST), p62 (1:1,000; CST), Smad3 (1:1,000; Abcam), p-Smad3 (1:1,000; CST), Ubiquitin (1:1,000; CST), HA (1:1,000; CST), ULK1 (1:1,000; CST), Beclin1 (1:1,000; CST), ATG3 (1: 1,000; CST), ATG4 (1:1,000; CST), ATG5 (1:1,000; CST), ATG7 (1:1,000; CST), ATG12 (1:1,000; CST), Nedd4L (1:1,000; Abclonal), STUB1 (1: 1,000; Abclonal), and VHL (1:1,000; Abclonal).

## RNA extraction and RT-qPCR analysis

Total RNA was extracted from cultured cells using Trizol reagent (Sangon Biotech) and RNA was reverse transcribed into cDNA using PrimeScrip RT reagent Kit (Takara), according to the manufacturer's recommendations. RT-qPCR reactions were performed by Takara's SYBR Premix Ex Taq II (Tli RNaseH Plus) in Applied Biosystems 7500 Fast Real-Time PCR System, with gene-specific primers which were shown in Table S5. Transcripts were normalized to GAPDH mRNA.

## Dual-luciferase reporter assay

H293T cells cultured in 24-well plates were transfected with experimental or control firefly reporter vectors and Renilla co-reporter vector. These cells were lysed with passive lysis buffer at 24 h after transfection, and the luciferase activity was detected according to the protocol of Dual-Luciferase Reporter Assay System (Promega). Constitutively activated Smad3-2SD plasmid was generated by using PCR-based mutagenesis to substitute Ser[423] and Ser[425] to aspartic acid in Smad3 and cloned into pcDNA3.1 vector.

## Chemosensitivity and CCK-8 assay

Transfected or CDDP-resistant or parental lung cancer cells were incubated into 96-well plates and treated with different concentration of CDDP, with or without CQ or 3-MA. After 48 h, cell viability was assessed by the absorbance (450 nm) using Cell Counting Kit-8 (CCK-8; NCM biotech) according to the manufacturer's protocol. The cell growth curve was drawn and the 50% inhibition of growth ($IC_{50}$) value of each cell line was calculated using GraphPad Prism7.0.

## Subcellular RNA fractionation

To determine the cellular localization of Linc00673-V3, the cytoplasm and nuclear RNA fractions were isolated and purified using PARIS system (Thermo Fisher Scientific), according to the manufacturer's protocol. Then the levels of Linc00673-V3, GAPDH, and U6 in the extracted RNAs were analyzed by RT-qPCR.

## FISH

FISH was performed according to the manufacturer's instructions (RNA FISH Kit; GenePharma). In brief, A549 and H1975 cells grown on the slides were washed with PBS twice and fixed in 4% paraformaldehyde. Then slides were incubated with prehybridization buffer at 37°C for 30 min. Biotin-labeled probes were premixed with streptavidin-labeled Cy3 to form the working solution. After prehybridization, slides were incubated with probe working solution at 37°C overnight. Images were captured using an Olympus FV1000 confocal microscope (Olympus) the next day.

## RAP assay

Affinity pulldown of biotinylated RNA for detection of protein–RNA complexes was performed using Ribo RNAmax-T7 biotin-labeled transcription kit (RiboBio), according to the manufacturer's protocol. In vitro transcription of the biotinylated RNA and two 10-cm dishes of ~70% confluent cells were prepared for the pulldown reaction, incubated at RT for 30 min. Washed beads were added to the RNA–protein mixture and incubated for another 30 min. With the supernatant discarded, 1x Laemmli sample buffer supplemented with $\beta$-mercaptoethanol was added to the magnetic beads for Western blotting.

## RIP and cross-linking RIP (CLIP) assay

RIP assay was performed using BersinBio RIP Kit, according to the manufacturer's protocol. The cultured cells were lysed by RIP lysis buffer containing protease inhibitor and RNase inhibitor, and the lysate was incubated with specific antibodies. The mixture was incubated with magnetic beads, and these beads were washed three times to obtain the co-immunoprecipitated RNAs, which were extracted by Trizol reagent later. The RNAs were subjected to RT-qPCR analysis, with IgG used as control.

CLIP assay was performed using BersinBio CLIP-qPCR Kit (BersinBio), according to the manufacturer's protocol. Briefly, cells were cultured in medium containing 100 $\mu$M 4-thiourdine overnight. The next day, ultraviolet irradiation was applied to cause covalent binding between RNA fragments and proteins. After harvesting the cells with NP-40 lysis buffer, RNase was applied to digest unbound RNA fragments and the targeted protein–RNA complex was captured using anti-Smad3 monoclonal antibody. The RNA fragments obtained by immunoprecipitation were subjected to reverse transcription and poly(A) tailing reaction at 3'OH ends of each RNA fragment. Then, RT-qPCR was conducted, primers were designed every 200 nt with 100 nt overlapping intervals to cover the full length of Linc00673-V3 as depicted in Fig 4G and detail sequences of each primer were listed in Table S2.

## ChIP assay

ChIP was performed using ChIP Pierce Magnetic ChIP Kit (Thermo Fisher Scientific) according to the manufacturer's protocol. Briefly, cells ($4 \times 10^6$ cells) were fixed and cross-linked, and the cross-linked cells were lysed by buffer containing protease/phosphatase inhibitors and digested by MNase. Supernatant containing the digested chromatin was incubated with anti–p-Smad3 antibody overnight at 4°C. The magnetic beads were added to bind the immunoprecipitation complex. Beads were washed and purified to obtain the DNA for RT-qPCR analysis. The anti-RNA polymerase II antibody was used as a positive control and the IgG was used as a negative control.

## Co-immunoprecipitation assay

The cultured cells were lysed by NP40 lysis buffer (Beyotime) containing protease inhibitor cocktail at 4°C for 45 min. Rabbit or mouse IgG and Protein A/G PLUS-Agarose (Santa Cruz Biotechnology) were added for pre-cleaning at 4°C for 30 min, and the cell lysate was incubated with primary antibody at 4°C overnight. The complex was added with Protein A/G PLUS-Agarose for co-immunoprecipitation, which was collected with sample buffer for Western blotting analysis.

### Immunofluorescent staining

Transfected or control cells were incubated on sterilized coverslips in six-well plates, with or without autophagy activator rapamycin. After 24 h, cells were fixed using 4% paraformaldehyde for 5 min at RT, and fixed cells were permeabilized with 1% Triton X-100 for 10 min. After blocked with 5% BSA in PBST for 30 min, cells were incubated in primary antibody against LC3B (1:200; Proteintech) at 4°C overnight and in fluorochrome-labeled anti-rabbit secondary antibody (1:500; Beyotime) for 1.5 h at RT in the dark. Afterwards, the coverslips were stained with DAPI (1:10,000; Beyotime) and sealed with anti-fluorescence quenching sealing liquid (Beyotime). Images were taken from Olympus FV1000 confocal microscope (Olympus).

### Animal experiment

BALB/c nude mice (6 wk old) were purchased from Shanghai SLAC Laboratory Animal Co., Ltd and sustained in the pathogen-free environment. The animal experiment protocol was approved by the Animal Care and Use Committee of the Laboratory Animal Center Zhejiang University.

A549 cells ($2 \times 10^6$) were resuspended in 0.9% saline with 0.1 ml Matrigel (Corning). Then, the cells were injected subcutaneously into the right flank of 6-wk-old female BALB/c nude mice. Mice bearing tumors of 150–200 mm$^3$ were randomly assigned into four groups: vehicle control group (saline), ASO-Linc00673-V3 group (5 nmol intratumoral injection), cisplatin group (3 mg/kg), and cisplatin (3 mg/kg) together with 5 nmol ASO-Linc00673-V3 group. The treated mice were intraperitoneally injected twice per week for 4 wk. Mice were euthanized for tumor dissection, immunohisto-chemistry staining, and RT-qPCR analysis. Tumor weight was measured at the end point of the experiment. Data were analyzed using $t$ test.

All tumor grafts were subjected to immunohistochemical staining according to the manufacturer's instructions (SP Rabbit &Mouse HRP Kit; CWBIO). In brief, paraffin-embedded tissue slices were dewaxed and rehydrated. After that, antigens were retrieved by boiling in 0.01 M citrate buffer for 30 min. Tissues were incubated with 3% hydrogen peroxide followed with goat serum. Antibodies used in immunohistochemical staining were LC3B (1:300; Proteintech) and Smad3 (1:300; Proteintech). DAB was applied to immunode-tection after incubating tissue slices with primary antibodies at 4°C overnight.

### Bioinformatics and statistical analysis

The secondary structure of Linc00673-V3 was predicted by bio-informatic tool RNAfold (http://rna.tbi.univie.ac.at). The Smad3-binding sites in *LC3B* promotor were putative using the bio-informatic tool JASPAR (Fornes et al, 2020) (https://jaspar.genereg.net). The correlation between expression levels of Linc00673-V3, Smad3, and *LC3B* (also known as MAP1LC3B) and overall survival outcomes was obtained from KM plotter (https://kmplot.com/analysis).

Statistical analyses were performed using Graphad Prism7.0. All experiment data were expressed as mean ± SD of at least three independent experiments, and the *P*-value was calculated by two-tailed $t$ test or analysis of variance. Unless specified, $P < 0.05$ was considered significant.

## Data Availability

All data that support the findings of this study are available from the corresponding authors upon reasonable request.

### Ethics statement

The animal experiment protocol was approved by the Animal Care and Use Committee of the Laboratory Animal Center Zhejiang University.

## Supplementary Information

## Acknowledgements

This work was supported by National Natural Science Foundation of China (grant numbers 32370584, 21976155); Zhejiang Provincial Natural Science Foundation of China (grant number LY18C06001); and CAMS Innovation Fund for Medical Sciences (CIFMS) (grant number: 2019-I2M-5-044); the Funda-mental Research Funds for the Central Universities (to Y Wu); and the Key R&D Program of Zhejiang (2023C03172) to J Xu. We thank Qiong Huang and Liyan Wang from the Core Facilities, Zhejiang University School of Medicine for their technical supports.

### Author Contributions

H Ni: conceptualization, methodology, and writing—original draft, review, and editing.
S Tang: methodology.
G Lu: validation and writing—original draft.
Y Niu: methodology.
J Xu: data curation.
H Zhang: resources and supervision.
J Hu: resources and supervision.
H-M Shen: resources and supervision.
Y Wu: conceptualization, visualization, and writing—original draft.
D Xia: conceptualization, resources, supervision, funding acquisi-tion, and writing—original draft.

### Conflict of Interest Statement

The authors declare that they have no conflict of interest.

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
