## [Reviewer comments · Life Science Alliance]

Life Science Alliance

Linc00673-V3 positively regulates autophagy by promoting Smad3 mediated LC3B transcription in NSCLC

Heng Ni, Song Tang, Guang Lu, Yuequn Niu, Jinming Xu, Honghe Zhang, Jian Hu, Han-Ming Shen, Yihua Wu, and Dajing Xia
DOI: <https://doi.org/10.26508/lsa.202302408>

Corresponding author(s): Dajing Xia, Zhejiang University and Yihua Wu, Zhejiang University

Review Timeline:

Submission Date:	2023-09-30
Editorial Decision:	2023-11-20
Revision Received:	2024-02-18
Editorial Decision:	2024-03-07
Revision Received:	2024-03-13
Accepted:	2024-03-13

Transaction Report:

November 20, 2023

Re: Life Science Alliance manuscript #LSA-2023-02408-T

Prof. Dajing Xia
Zhejiang University
School of Public Health
Hang Zhou
China

Dear Dr. Xia,

Thank you for submitting your manuscript entitled "Linc00673-V3 positively regulates autophagy by promoting Smad3 mediated LC3B transcription in NSCLC" to Life Science Alliance. The manuscript was assessed by expert reviewers, whose comments are appended to this letter. We invite you to submit a revised manuscript addressing the Reviewer comments.

Thank you for this interesting contribution to Life Science Alliance. We are looking forward to receiving your revised manuscript.

Sincerely,

B. MANUSCRIPT ORGANIZATION AND FORMATTING:

Reviewer #1 (Comments to the Authors (Required)):

In the current manuscript Heng Ni and colleagues investigated the role of the long non-coding RNA LINC00673 in cisplatin-resistance and autophagy in lung cancer. The authors describe a chemoresistance-promoting role of isoform 3 (V3) of LINC00673 and link this to its positive effect on autophagy by enhancing LCB3 gene expression. Mechanistically, the authors show that LINC00673-V3 can bind to Smad3. This interaction seems to prevent the ubiquitination of Smad3 by the E3 ligase STUB1 leading to an accumulation of Smad3 in LINC00673 overexpression. Smad3 in turn can initiate the transcription of LC3B thereby enhancing the autophagy and chemoresistance of the cells. The authors furthermore provide in vivo data that suggest that inhibition of LINC00673-V3 using ASOs might be a valid anti-cancer therapy and could synergize with cisplatin treatment.

Overall, the manuscript follows a logical flow and the authors present a large collection of data that significantly expand our knowledge about the function of LINC00673 as well as the regulation of autophagy by this ncRNA in conjunction with Smad3. However, some effect sizes are rather small and individual western blots would benefit from a quantification. In addition, some language editing should be considered. Please find my additional comments below:

- 1) The authors mention that p62 is degraded faster/more in cisplatin-resistant cells (Fig.1C) or after LINC00673-V3 overexpression. However, the authors only present a western blot showing the steady state level of p62. Thus, "degradation" was not measured/analyzed. Also, the knockdown of LINC00673 does not seem to affect p62 levels in a consistent manner. Please comment.
- 2) Please provide experimental proof for efficient and specific Smad3 immunoprecipitation (Fig.4).
- 3) The FISH analysis shown in Figure 4 requires a specificity control, e.g. cells with KD/KO of LINC00673.
- 4) Is Smad3 protein increased in A549/CDDP cells and after cisplatin treatment of parental cells?
- 5) Please show that overexpression of LINC00673-V3_full length but not the truncated versions (Fig. 4I) can stabilize Smad3 protein in A549 cells.
- 6) In line with the comment above, please show that overexpression of LINC00673-V3 and its truncated versions can affect (or not) the interaction between Smad3 and STUB1 proteins.
- 7) Please show consistently total and p-Smad3 expression in similar assays.
- 8) There is no difference in Smad3 and p-Smad3 levels between si-NC and si-LINC00673-V3 in Figure 5D and 5E (compare with Figure 5C). Please comment.
- 9) Please use the same cisplatin concentration range and x-axis formatting throughout the manuscript (see Figure 2 and Figure 7 - mix between log and linear scale)
- 10) Please explain the autophagy reporter (sens-GFP-LC3 and stub-RFP-LC3) in more detail, e.g. in the material and methods section.
- 11) The authors should introduce the scheme of the different LINC00673 isoforms/variants shown in Figure S4A earlier in the text/figures.
- 12) The current manuscript version contains a wrong figure legend for Figure S3. Also, Figure S6E and S6F contain a wrong sample label. What is quantified in S6A and S6B (mRNA or protein)? Please specify!
- 13) In Figure 6C, please indicate what has been detected (e.g. axis label with LC3B mRNA).
- 14) Please include clone/catalogue numbers for all antibodies used in this study.

Reviewer #2 (Comments to the Authors (Required)):

The lncRNA linc00673 (also known as SLNCR1) has been implicated in multiple different cancers. In this manuscript the authors show that a specific isoform of linc00673, termed linc00673-V3, is induced upon cisplatin treatment in NSCLC. The work proposes that linc00673-V3 binds and regulates the stability of Smad3 to promote autophagy and chemoresistance. This is an interesting manuscript proposing a function and mechanism of action for linc00673-V3 in NSCLC. However, the conclusions are not fully supported by the data. There are a number of technical concerns that dampen by enthusiasm for this study and would need to be addressed. See below:

Fig 1A: Do GAPDH levels change in cisplatin resistant cells? Is this a good ref gene?

Fig 1G-I: linc00673-V1 isoform is also upregulated in these experiments. Have the authors tested it's involved in cisplatin resistance? If not, why was this transcript dropped from the analysis?

Fig 2B-F: The authors should confirm that the siRNAs used are specific for linc00673-v3 isoform and don't affect expression of any of the others? In particular, the v1 isoform.

Fig 2B, C, Fig S2 (and Fig 3D, 5A, B): I struggle to ascertain whether the 150-fold overexpression of linc00673-v3 used in these experiments is biologically relevant. The authors may be better off using a different promoter construct to up-regulate linc00673-v3 by a smaller amount or attempt CRISPRa.

Fig 3: the panels are very small and it is very hard to actually visualise the data.

Fig 3F: are the different isoforms expressed to similar levels as linc00673-v3 to allow for comparison? The western blot data should be quantified.

The authors state that linc00673-V3 may promote LC3B transcription as they are both induced by cisplatin. No direct evidence for this is shown and the sentence should be toned down.

Fig 4A: The authors state that the bands around 55kDa look significantly different. I disagree with this interpretation as it looks to me like there is higher conc of interacting proteins in sense pull down compared to control and that there are no obvious bands present in one sample compared to the other from looking at the silver stain. This sentence could be re-phrased.

Fig 4G, H: has the primer efficiency been determined for each primer pair to confirm that they all amplify template similarly?

Fig 4I: do the small fragments on their own bind Smad3?

Fig 5D-H: linc00673-V3 kd studies are carried out using a single siRNA. Another siRNA should be used to deplete this transcript in these experiments to control for off-target effects (in fact all siRNA kd experiments should be carried out with 2 independent constructs).

Fig 7C-E: a control region close to regions a and b that is not predicted to bind Smad3 should be used to examine specificity of the interaction of Smad3 with regions a and b within the linc00673 promoter.

Fig 7M-O: again 2 siRNAs should be used.

Fig 8: again 2 ASOs should be used to control for off-target effects.

Fig 8J: the authors should show whether linc00673-v3 kd also reduces the growth of NSCLC cells or whether the effect is specific to cisplatin treatment.

Further points:

Grammar should be checked throughout the manuscript.

Can the authors describe the stubRFP-sensGFP-LC3 reporter? I couldn't work out how this works and an explanation would help with understanding.

Reviewer #3 (Comments to the Authors (Required)):

This is an interesting article that identifies linc00673-V3 as a positive regulator of autophagy in NSCLC by promoting Smad3-

mediated LC3B transcription. The involvement of lncRNAs in the molecular regulation of autophagy and their downstream targeted cancer-related properties such as chemoresistance is quite novel.

As a general comment, although the manuscript includes an extremely large volume of data, the presented data flow is very complicated and hard to follow, thus resulting in lack of focus. This weakness is further enhanced by the observed inconsistency regarding the experimental models (cell lines) that the authors use throughout the ms to validate their hypotheses and conclusions. For example, the first experiments (referring to those presented in figures 1 and 2) have been conducted in A549, H1975, and A549/CDDP cells, while the rest of the experiments only performed in A549 and H1975 parental cell lines, although the focus of the study is the chemoresistance.

Specific comments

- The involvement of LC3B/ P62 in autophagy, especially that of the LC3BI/II isoforms, should be explained more thoroughly in the introduction
- The authors need to clarify the cell culture conditions of A549/CDDP cells (CDDP resistant cells). Did they continuously expose the A549/CDDP cells in CDDP, for maintaining the resistant cell phenotype? If not, the authors should test whether the addition of CDDP in the resistant under the same experimental conditions may result in further differentiation in LC3B/P62 levels, as observed with the parental A549 and H1975 cells when treated with CDDP.
- A549 and H1975 cell line clearly differ in their responses to CDDP treatment. Do they differ in their aggressiveness, metastatic potential etc? An explanation should be given
- Some of the representative western blots presented in the figures fail to show significant differences before and after treatments (siRNA, CDDP etc) or among different cell lines. In these cases the authors should either replace the blots with clearer ones or provide corresponding graphs with the protein mean values after densitometry.
- The protein levels of LC3B I/II in either A549 or H1975 cells under the same treatment conditions are not consistent throughout the manuscript. For example, in Figure 3 A panel B the protein levels are not consistent with Figure 1 D panel B under the same treatment.
- In Figure 1 F the images of the controls are bluer, please use better images. Also, the scale needs to be more obvious.
- In Figure 1 H the control A549-NC, should be A549-DMSO like in Figure 1 I.
- Figure 2 A needs to be moved to Figure 1. It contains important information for the flow of the rest of the experiments.
- Figure 2 panels C,D,E,F,G,H and I should be consistent in presenting either the log concentration of CDDP or CDDP μM .
- Figure 3: In addition to parental lines, all the experiments shown in different panels should be also performed in CDDP resistant cells as a direct validation of the hypothesis
- Figure 3 A needs to be supplemented with a densitometry graph, like the one shown in Figure 3 B.
- Figure 3 C: The outcome of the experiment is not clear. Isn't the CQ an inhibitor of autophagy? Why upon CQ treatment there is an increase in LC3B levels? Explanation is necessary.
- Figure 3 D isn't that useful, because the information is already provided in Figure 3 F.
- Figure 3 F: The experimental data from the over-expression of the different isoforms alone (controls), are missing.
- Please provide protein quantification for Figure 5 G/H
- Figure 6 F/G: Over-expression of STUB1 results in lower Ub-Smad3 expression, while Linc00673-V3 over-expression reveals higher Ub-Smad3. This is in contrast to what the authors claim.
- Figure 8 K/L: The comparison between CDDP/ASO and ASO is lacking.
- In the discussion section the authors need to discuss more thoroughly their findings on the basis of the current status of knowledge regarding the role of lncRNAs in autophagy in cancer in general and in NSCLC specifically.

Reviewer #1

1) The authors mention that p62 is degraded faster/more in cisplatin-resistant cells (Fig.1C) or after LINC00673-V3 overexpression. However, the authors only present a western blot showing the steady state level of p62. Thus, "degradation" was not measured/analyzed. Also, the knockdown of LINC00673 does not seem to affect p62 levels in a consistent manner. Please comment.

Thank you for your suggestion. In Fig. 1C, we compared the autophagic levels between wild-type A549 cells and A549 cisplatin-resistant cells. We found that the p62 protein level was lower in the resistant cells, while the LC3B-II protein level was higher, suggesting a higher level of autophagy in the resistant cells. "Degradation" is often used to describe a dynamic process, but since we did not dynamically measure the protein level of p62 here, based on your suggestion, we have made a correction in the manuscript, replacing "degradation" with "decreased p62 protein level" (Line 183 in the revised manuscript).

Regarding the inconsistent effects on intracellular p62 protein levels after knocking down Linc00673, we believe the following: Autophagy itself is a dynamic process, where p62 plays a role in recruiting cargo to the phagophore. There exists a dynamic equilibrium between the generation and degradation of p62 through the autophagic pathway. Upon stimulation, such as enhancement of autophagy in the early stage, there is an increase in p62 transcription and protein levels. As the autophagic process proceeds, p62 undergoes degradation via autolysosomes, resulting in a gradual decrease in its protein level^{1,2}. In this study, we found that knocking down Linc00673-V3 primarily affects cellular autophagy by downregulating the expression level of LC3B. When the intracellular level of LC3B protein decreases, its impact on the p62 protein level is a complex dynamic process. On one hand, when the LC3B protein level is reduced and autophagic flux is inhibited, it can result in the accumulation of p62 protein (Fig 8F). However, in the early stages, before significant changes in autophagic flux occur, the protein level of p62 may not show significant changes (Fig 3A, 3C). In this study, we focus more on the direct regulatory effect of Linc00673-V3 on LC3B. Moreover, considering the overall trend of LC3B protein level changes, knocking down Linc00673 generally exerts an inhibitory effect on the cellular autophagic process.

2) Please provide experimental proof for efficient and specific Smad3 immunoprecipitation (Fig.4).

Thank you for your suggestion. Following your advice, we conducted Western blot experiments to demonstrate the efficiency of Smad3 immunoprecipitation in RIP and CLIP assays.

[Figure removed by editorial staff per authors' request].

3) The FISH analysis shown in Figure 4 requires a specificity control, e.g. cells with KD/KO of LINC00673.

Thank you for your suggestion. Following your advice, we knocked down the expression level of linc00673-V3 in A549 cells. FISH results showed a significant decrease in the red fluorescence intensity of Linc00673-V3. Concurrently, there was also a decrease in the fluorescence intensity of Smad3, suggesting a decrease in the Smad3 protein level. However, knocking down the expression level of Linc00673-V3 did not affect its colocalization with Smad3.

[Figure removed by editorial staff per authors' request].

A549 (si-Linc00673-V3)

4) Is Smad3 protein increased in A549/CDDP cells and after cisplatin treatment of parental cells?

Thank you for your suggestion. Following your advice, we examined the protein expression levels through Western blotting and found that the Smad3 protein level increased in A549/CDDP cells and parental cells treated with cisplatin. The relevant experimental results are as follows and are added to Fig. 7K and 7L.

[Figure removed by editorial staff per authors' request].

5) Please show that overexpression of LINC00673-V3_full length but not the truncated versions (Fig. 4I) can stabilize Smad3 protein in A549 cells.

Thanks for your suggestion, as demonstrated in the panel below, only overexpression of Linc00673-V3 full length but not the truncations could stabilize Smad3 protein.

[Figure removed by editorial staff per authors' request].

6) In line with the comment above, please show that overexpression of LINC00673-V3 and its truncated versions can affect (or not) the interaction between Smad3 and STUB1 proteins.

Thanks for your suggestion, as demonstrated in the panel below, only overexpression of

Linc00673-V3 full length but not the truncations could affect the interaction between Smad3 and STUB1.

A

[Figure removed by editorial staff per authors' request].

7) Please show consistently total and p-Smad3 expression in similar assays.

Thank you for your suggestion. In order to maintain consistency in the detection of Smad3 and p-Smad3, we have supplemented the detection of Smad3 expression levels in the figures of Fig. 7K and 7L.

[Figure removed by editorial staff per authors' request].

8) There is no difference in Smad3 and p-Smad3 levels between si-NC and si-LINC00673-V3 in Figure 5D and 5E (compare with Figure 5C). Please comment.

Thank you for your suggestion. Fig 5C depicts the detection of Smad3 protein levels at 24h and 48h time points under treatment with si-Linc00673 alone. Meanwhile, Fig 5D and 5E were conducted to demonstrate that the decrease in Smad3 protein levels induced by si-Linc00673 is due to increased Smad3 degradation. Therefore, cycloheximide (CHX) was used to inhibit protein synthesis in the experiments of Fig 5D and 5E. It is essential to ensure that the baseline levels of Smad3 in the NC group and the si group treated with CHX are equivalent at the 0h point. Thus, the 0h CHX treatment corresponds to the 12h time point of siRNA treatment. At this point, due to the short duration of siRNA treatment, there is no difference in Smad3 protein levels between the si-NC and si-LINC00673 groups. However, with prolonged siRNA treatment and under CHX-mediated inhibition of protein synthesis, the difference in Smad3 protein levels between the two groups becomes gradually evident.

9) Please use the same cisplatin concentration range and x-axis formatting throughout the manuscript (see Figure 2 and Figure 7 - mix between log and linear scale)

Thank you for your suggestion. Following your advice, we have reprocessed the data for the H1299, H596, and A549 cell lines in Figure 2 and Figure 7, and unified the x-axis formatting to a log scale. The results are as follows:

[Figure removed by editorial staff per authors' request].

10) Please explain the autophagy reporter (sens-GFP-LC3 and stub-RFP-LC3) in more detail, e.g. in the material and methods section.

Thank you for your suggestion. Following your advice, we explained the stubRFP-sensGFP-LC3 plasmid in the material and methods section (Line 558-564).

In detail, in the stubRFP-sensGFP-LC3 plasmid, the fusion protein components include the red fluorescent protein stubRFP, the green fluorescent protein sensGFP, and the autophagy marker protein LC3. SensGFP is an acid-sensitive protein, while stubRFP is a stable fluorescent protein unaffected by pH changes. During the initial stages of autophagy, cytoplasmic LC3 transitions to membrane-associated LC3, guiding the aggregation of stubRFP-sensGFP-LC3 onto autophagosomes. This can be observed as punctate red/green colocalized aggregates under fluorescence microscopy. In the later stages of autophagy, autophagosomes fuse with lysosomes to form autolysosomes, altering the environmental pH. When the pH is below 5, sensGFP fluorescence is quenched, and only red fluorescence punctate aggregates can be detected. Therefore, a decrease in sensGFP intensity indicates the efficiency of autolysosome formation. A lower sensGFP signal suggests smoother trafficking from autophagosomes to autolysosomes, whereas a higher signal indicates inhibition of autophagosome-lysosome fusion, impeding autolysosome formation. Since stubRFP is unaffected by acidic environments, the ratio of sensGFP to stubRFP bright spots can be used to evaluate the progression of autophagic flux.

11) The authors should introduce the scheme of the different LINC00673 isoforms/variants shown in Figure S4A earlier in the text/figures.

Thank you for your suggestion. Following your advice, we rearranged the Figure S4A to Figure S1G, and added the introduction of the scheme of the different LINC00673 isoforms earlier in the text (Line 190).

12) The current manuscript version contains a wrong figure legend for Figure S3. Also, Figure S6E and S6F contain a wrong sample label. What is quantified in S6A and S6B

(mRNA or protein)? Please specify!

Thank you for your suggestion. Following your advice, we have reviewed and corrected the legend for Figure S3 in the supplementary materials. Additionally, we have rectified the sample labels in Figure S6E and S6F. In Figures S6A and S6B, we have added that the y-axis represents the change in mRNA levels, clarifying the detection target.

13) In Figure 6C, please indicate what has been detected (e.g. axis label with LC3B mRNA).

Thank you for your suggestion. Following your advice, we have supplemented the vertical axis of Figure 6C to indicate that the detection target is LC3B mRNA.

14) Please include clone/catalogue numbers for all antibodies used in this study.

Thank you for your suggestion. Following your advice, we have supplemented the antibody information (catalogue numbers) in Supplementary Table 5.

Reviewer #2

1) Fig 1A: Do GAPDH levels change in cisplatin resistant cells? Is this a good ref gene?

Thank you for your suggestions. GAPDH, as a housekeeping gene, has been widely used as an internal reference in experiments. However, recent studies have found that GAPDH exhibits varying expression levels in different tumor cell lines. Moreover, research suggests that GAPDH may be involved in tumor progression and the regulation of tumor cell fate^{3,4}. Additionally, the expression level of GAPDH may be influenced by factors such as insulin, HIF-1, or nitric oxide (NO)³. Nevertheless, there is currently no research reporting a correlation between the expression level of GAPDH and drug resistance in tumor cells. Furthermore, many studies investigating the relationship between autophagy and tumor drug resistance have adopted GAPDH as an internal reference⁵⁻⁷. In addition, following your suggestion, we have also examined the expression levels of GAPDH and β -actin in both A549 and A549 CDDP resistant cells, finding no difference between them. Therefore, we believe that GAPDH does not play a role in chemoresistance in this study, and thus it can still be used as a reference gene.

[Figure removed by editorial staff per authors' request].

2) Fig 1G-I: linc00673-V1 isoform is also upregulated in these experiments. Have the authors tested it's involved in cisplatin resistance? If not, why was this transcript dropped from the analysis?

Thank you for your suggestion. We believe that although the expression level of Linc00673-V1 also increases, in Figure 3F, through rescue experiments, we found that only overexpression of Linc00673-V3 can rescue the expression level of LC3B. This evidence supports the notion that Linc00673-V3 plays the primary role. Additionally, consistent with previous research findings⁸, in NSCLC, the expression levels of Linc00673-V3 and V4 isoforms are predominant. Therefore, we conclude that compared to the V1 isoform, the V3 isoform exerts the main biological function.

3) Fig 2B-F: The authors should confirm that the siRNAs used are specific for linc00673-v3 isoform and don't affect expression of any of the others? In particular, the v1 isoform.

Thank you for your suggestion. We have assessed the expression levels of various isoforms after si-Linc00673-V3 treatment and found that although there is a slight decrease in the expression level of the V1 isoform, the most significant decrease is observed in the expression level of V3. Combining this with the results of the rescue experiment in Figure 3F, we believe that the primary impact after siRNA treatment is on the expression level of V3, consequently affecting the autophagic levels within the cells.

[Figure removed by editorial staff per authors' request].

4) Fig 2B, C, Fig S2 (and Fig 3D, 5A, B): I struggle to ascertain whether the 150-fold overexpression of linc00673-v3 used in these experiments is biologically relevant. The authors may be better off using a different promoter construct to up-regulate linc00673-v3 by a smaller amount or attempt CRISPRa.

Thank you for your suggestion. In Figure S2, we demonstrate a 150-fold overexpression detected

after overexpressing Linc673-V3 in H1299 cells. We believe that the partial reason for this exceptionally high level of overexpression is that Linc00673-V3 is barely expressed in H1299 cells (Figure 2A), leading to a somewhat exaggerated fold change upon overexpression. Subsequently, we examined the efficiency of overexpressing linc00673-V3 in A549 and H1975 cell lines. The overexpression efficiency in A549 cells was observed to be 40-fold, while in H1975 cells, it was 30-fold. We consider this level of overexpression to be biologically significant.

[Figure removed by editorial staff per authors' request].

5) Fig 3: the panels are very small and it is very hard to actually visualise the data.

Thank you for your suggestion. We have reorganized Figure 3 according to your suggestions, and we hope this revision addresses the issue effectively.

6) Fig 3F: are the different isoforms expressed to similar levels as linc00673-v3 to allow for comparison? The western blot data should be quantified.

Thank you for your suggestion. The expression levels of different isoforms vary under basal conditions. However, in Figure 3F, our intention was to explore the effects of different isoforms on LC3B protein levels through rescue experiments. Therefore, after knocking down the endogenous levels of Linc00673 in cells, we separately overexpressed different isoforms and compared their effects on LC3B protein levels with the control group. Additionally, following your suggestion, we quantified the protein levels, and the results of protein quantification can be found in Supplementary Figure S3D.

[Figure removed by editorial staff per authors' request].

7) The authors state that linc00673-V3 may promote LC3B transcription as they are both induced by cisplatin. No direct evidence for this is shown and the sentence should be toned down.

Thank you for your suggestion. We did identify some omissions in the original manuscript. Following your advice, we have made the necessary corrections to the original manuscript (Line 272).

8) Fig 4A: The authors state that the bands around 55kDa look significantly different. I disagree with this interpretation as it looks to me like there is higher conc of interacting proteins in sense pull down compared to control and that there are no obvious bands present in one sample compared to the other from looking at the silver stain. This sentence could be re-phrased.

[Figure removed by editorial staff per authors' request].

Thank you for your suggestion. We have carefully reviewed your advice and made modifications to the wording in the manuscript (Line 280). In the RNA Affinity Pulldown experiment, the "sense" group refers to the sense strand of the lncRNA transcribed in vitro, while the "antisense" group refers to the antisense strand of the transcribed lncRNA, which is typically used as an experimental negative control. In our results, the silver staining intensity near the 55 kDa molecular weight range was higher in the sense group compared to the control group (antisense). Referring to studies on similar RNA Affinity Pulldown experiments⁸⁻¹⁰, we believe that there are differences between the experimental and control groups in the results.

9)Fig 4G, H: has the primer efficiency been determined for each primer pair to confirm that they all amplify template similarly?

Thank you for your suggestion. Prior to conducting the CLIP experiment, we consulted numerous

similar articles⁹⁻¹¹. To ensure qPCR detection covers the entire length of the lncRNA as much as possible, primer sequences need to be designed at fixed intervals, which inevitably leads to variations in amplification efficiency among different primers. However, on the other hand, qPCR amplification results are also influenced by the complexity of the template. In the CLIP experiment, preceding steps such as immunoprecipitation and washing off nonspecifically bound RNA significantly reduce the complexity of the qPCR template, thereby minimizing the possibility of nonspecific amplification. Additionally, we utilized two different cell lines for replicates, further demonstrating the stability of the experimental results.

10) Fig 4I: do the small fragments on their own bind Smad3?

Thank you for your suggestion. We have detected the binding ability of the two small fragments (1-100bp, 2200-2275bp) and found that they cannot directly bind to Smad3.

[Figure removed by editorial staff per authors' request].

11) Fig 5D-H: linc00673-V3 kd studies are carried out using a single siRNA. Another siRNA should be used to deplete this transcript in these experiments to control for off-target effects (in fact all siRNA kd experiments should be carried out with 2 independent constructs).

Thank you for your suggestion. Following your advice, we repeated the experiment using a new siRNA and found that the results remained consistent with the conclusions presented in the manuscript. Details are as follows:

[Figure removed by editorial staff per authors' request].

12) Fig 7C-E: a control region close to regions a and b that is not predicted to bind Smad3 should be used to examine specificity of the interaction of Smad3 with regions a and b within the linc00673 promoter.

Thank you for your suggestion. Following your advice, we selected the (c) region outside of the predicted (a) and (b) regions and conducted ChIP experiments on it. The experimental results indicated that Smad3 did not show significant binding ability to the (c) region. This further confirms the specificity of Smad3 binding to the (b) region. Details are presented in supplementary Fig. S7D, S7E.

[Figure removed by editorial staff per authors' request].

13) Fig 7M-O: again 2 siRNAs should be used.

Thank you for your suggestion. Following your advice, we added an additional siRNA targeting Smad3 and repeated the experiments shown in Fig. 7M-O. The experimental results are as follows:

[Figure removed by editorial staff per authors' request].

14) Fig 8: again 2 ASOs should be used to control for off-target effects.

Thank you for your suggestion. We have ordered a new ASO and validated it in vitro cell experiments. However, adding just one additional ASO replicate to the existing animal experiment design was considered a violation of the reduction principle of the 3Rs in animal experiments¹². Consequently, our application for animal experiments was not approved by the Zhejiang University Animal Experiment Ethics Committee. However, based on the existing in vitro experimental results, it is evident that the ASO targeting Linc00673-V3 can significantly inhibit cellular autophagy levels.

[Figure removed by editorial staff per authors' request].

15) Fig 8J: the authors should show whether linc00673-v3 kd also reduces the growth of NSCLC cells or whether the effect is specific to cisplatin treatment.

[Figure removed by editorial staff per authors' request].

Thank you for your suggestion. In our preliminary work, we indeed observed that knocking down Linc00673-V3 suppressed tumor growth *in vivo*. Therefore, in subsequent experimental designs, we first established subcutaneous tumors and then knocked down the expression of Linc00673-V3 using ASO after the tumors reached a similar size, aiming to minimize the impact of Linc00673-V3 on tumor growth. Consequently, in Figure 8J of the manuscript, our results mainly focus on the effect of Linc00673-V3 on cisplatin treatment.

Further points:

16) Grammar should be checked throughout the manuscript.

Thank you for your suggestion. During the revision process of the manuscript, we also sought help from a native speaker to assist in editing the grammar of the article. Hopefully this issue can be improved.

17) Can the authors describe the stubRFP-sensGFP-LC3 reporter? I couldn't work out how

this works and an explanation would help with understanding.

Thank you for your suggestion. Following your advice, we explained the stubRFP-sensGFP-LC3 plasmid in the material and methods section (Line 558-564).

In detail, in the stubRFP-sensGFP-LC3 plasmid, the fusion protein components include the red fluorescent protein stubRFP, the green fluorescent protein sensGFP, and the autophagy marker protein LC3. SensGFP is an acid-sensitive protein, while stubRFP is a stable fluorescent protein unaffected by pH changes. During the initial stages of autophagy, cytoplasmic LC3 transitions to membrane-associated LC3, guiding the aggregation of stubRFP-sensGFP-LC3 onto autophagosomes. This can be observed as punctate red/green colocalized aggregates under fluorescence microscopy. In the later stages of autophagy, autophagosomes fuse with lysosomes to form autolysosomes, altering the environmental pH. When the pH is below 5, sensGFP fluorescence is quenched, and only red fluorescence punctate aggregates can be detected. Therefore, a decrease in sensGFP intensity indicates the efficiency of autolysosome formation. A lower sensGFP signal suggests smoother trafficking from autophagosomes to autolysosomes, whereas a higher signal indicates inhibition of autophagosome-lysosome fusion, impeding autolysosome formation. Since stubRFP is unaffected by acidic environments, the ratio of sensGFP to stubRFP bright spots can be used to evaluate the progression of autophagic flux.

Reviewer #3

As a general comment, although the manuscript includes an extremely large volume of data, the presented data flow is very complicated and hard to follow, thus resulting in lack of focus. This weakness is further enhanced by the observed inconsistency regarding the experimental models (cell lines) that the authors use throughout the ms to validate their hypotheses and conclusions. For example, the first experiments (referring to those presented in figures 1 and 2) have been conducted in A549, H1975, and A549/CDDP cells, while the rest of the experiments only performed in A549 and H1975 parental cell lines, although the focus of the study is the chemoresistance.

1) The involvement of LC3B/ P62 in autophagy, especially that of the LC3BI/II isoforms, should be explained more thoroughly in the introduction

Thank you for your suggestion. Following your advice, we have added introductions to LC3BI/II transformation in the introduction section of the article (Line 112-118).

2) The authors need to clarify the cell culture conditions of A549/CDDP cells (CDDP resistant cells). Did they continuously expose the A549/CDDP cells in CDDP, for maintaining the resistant cell phenotype? If not, the authors should test whether the addition of CDDP in the resistant under the same experimental conditions may results in further differentiation in LC3B/P62 levels, as observed with the parental A549 and H1975 cells when treated with CDDP.

Thank you for your suggestion. Following your advice, we have supplemented the description of

the A549/CDDP cell culture conditions in the Materials and Methods section (Line 538-540). The A549/CDDP cells were purchased from the Cell Resource Center, Peking Union Medical College (PCRC), China. During culturing, the A549/CDDP cells were continuously exposed to 20 μ M concentration of CDDP to maintain their resistance to cisplatin. Therefore, in the experimental design, we did not measure the levels of LC3B/P62 in A549-resistant cells under CDDP treatment.

3) A549 and H1975 cell line clearly differ in their responses to CDDP treatment. Do they differ in their aggressiveness, metastatic potential etc? An explanation should be given

Thank you for your suggestion. In our previous work (Fig 2h, I from the reference 12)¹³, A549 and H1975 did not exhibit significant differences in migration and invasion abilities, as shown in the figure below. In this study, we examined the IC₅₀ values of A549 and H1975 for CDDP treatment and found no significant differences between them (Fig 2E, F in the manuscript), indicating that they do not differ significantly in sensitivity to CDDP treatment. The differences you mentioned between them under CDDP treatment may be attributed to their different levels of autophagy. We believe that intracellular autophagy is a complex dynamic process influenced by many factors, including cell status, cell growth density, etc., and differences in baseline autophagy levels may also exist among different cells. However, overall, the trend indicates that CDDP treatment upregulates the overall autophagy levels in both cell types.

[Figure removed by editorial staff per authors' request].

4) Some of the representative western blots presented in the figures fail to show significant differences before and after treatments (siRNA, CDDP etc) or among different cell lines. In these cases the authors should either replace the blots with clearer ones or provide corresponding graphs with the protein mean values after densitometry.

Thank you for your suggestion. We have replaced the figures in the original manuscript where the trend was not significant (Fig. 1D, Fig. 7K, 7L).

5) The protein levels of LC3B I/II in either A549 or H1975 cells under the same treatment conditions are not consistent throughout the manuscript. For example, in Figure 3 A panel B the protein levels are not consistent with Figure 1 D panel B under the same treatment.

Thank you for your suggestion. During the process of cellular autophagy, LC3B undergoes a conversion from LC3B-I to LC3B-II. Initially, LC3B-I, synthesized from mRNA translation, exists in its unmodified form within the cell. However, upon initiation of autophagy, it undergoes enzymatic cleavage, resulting in a lipidated form known as LC3B-II. LC3B-II is a critical component of the autophagy process; it binds to the inner membrane of autophagosomes and participates in transporting cellular components destined for degradation to lysosomes. This process is considered a hallmark of the execution phase of autophagy. In the assessment of autophagy levels, the conversion from LC3B-I to LC3B-II is widely used as an indicator of autophagic activity because the appearance of LC3B-II is often associated with the formation of autophagosomes and an increase in autophagic activity. However, as mentioned earlier, the baseline autophagy levels within cells are influenced by various external factors, making it difficult to ensure that cells in different experiments are at the same baseline autophagy level. Therefore, inconsistencies in baseline levels of LC3B I/II between different experiments may occur. So, in autophagy research, it is more common to compare changes in LC3B I/II within the same experimental group before and after treatment to assess the impact of the treatment on autophagy levels^{14, 15}.

6) In Figure 1 F the images of the controls are bluer, please use better images. Also, the scale needs to be more obvious.

Thank you for your suggestion. We have adjusted the font size of the scale to make it clearer. Additionally, regarding the issue you mentioned about the images of the controls being bluer (greener), the explanation is as follows: In Fig. 1E and Fig. 1F, we used the stubRFP-sensGFP-LC3 plasmid to indicate the status of autophagic flux. In the stubRFP-sensGFP-LC3 plasmid, the fusion protein components include the red fluorescent protein stubRFP, the green fluorescent protein sensGFP, and the autophagy marker protein LC3. SensGFP is an acid-sensitive protein, while stubRFP is a stable fluorescent protein unaffected by pH changes. During the initial stages of autophagy, cytoplasmic LC3 transitions to membrane-associated LC3, guiding the aggregation of stubRFP-sensGFP-LC3 onto autophagosomes. This can be observed as punctate red/green colocalized aggregates under fluorescence microscopy. In the later stages of autophagy, autophagosomes fuse with lysosomes to form autolysosomes, altering the environmental pH. When the pH is below 5, sensGFP fluorescence is quenched, and only red fluorescence punctate aggregates can be detected. Therefore, a decrease in sensGFP intensity indicates the efficiency of autolysosome formation. A lower sensGFP signal suggests smoother trafficking from autophagosomes to autolysosomes. Therefore, observing a lower green fluorescence signal after CDDP treatment indicates the progression of autophagic flux, consistent with the experimental expectations.

7) In Figure 1 H the control A549-NC, should be A549-DMSO like in Figure 1 I.

Thank you for your suggestion. We have made the correction to the image Figure 1H.

8) Figure 2 A needs to be moved to Figure 1. It contains important information for the flow of the rest of the experiments.

Thank you for your suggestion. After careful consideration of your advice, we have decided to retain Fig 2A. In Figure 1, the focus is primarily on observing higher levels of Linc00673 expression and autophagy after chemotherapy drug treatment, without comparing between different cell lines. However, in Figure 2, to further investigate the impact of Linc00673-V3 expression levels on cellular sensitivity to cisplatin treatment, it's necessary to determine the expression levels of Linc00673-V3 in different cell lines before conducting knockdown or overexpression experiments. Therefore, including Fig 2A in Figure 2 appears to be more appropriate. Thank you for considering our suggestion.

9) Figure 2 panels C,D,E,F,G,H and I should be consistent in presenting either the log concentration of CDDP or CDDP μ M.

Thank you for your suggestion. Following your advice, we have reprocessed the data for the H1299, H596, and A549 cell lines in Figure 2 and Figure 7, and unified the x-axis formatting to a log scale. The results are as follows:

[Figure removed by editorial staff per authors' request].

10) Figure 3: In addition to parental lines, all the experiments shown in different panels should be also performed in CDDP resistant cells as a direct validation of the hypothesis

Thank you for your suggestion. We have validated the experimental results in A549/CDDP cells and have supplemented them in Supplementary Figure 3.

[Figure removed by editorial staff per authors' request].

11) Figure 3 A needs to be supplemented with a densitometry graph, like the one shown in Figure 3 B.

Thank you for your suggestion. We have added the densitometry graph for Fig. 3A to supplementary Figure 3A.

[Figures removed by editorial staff per authors' request].

12) Figure 3 C: The outcome of the experiment is not clear. Isn't the CQ an inhibitor of autophagy? Why upon CQ treatment there is an increase in LC3B levels? Explanation is necessary.

Thank you for your suggestion. During the process of autophagy, autophagosomes fuse with lysosomes to form autolysosomes, ultimately leading to the degradation of their contents. Autolysosomes require an acidic environment to maintain the activity of relevant enzymes. CQ (Chloroquine) increases the pH of autolysosomes, thereby inhibiting their degradative function.

LC3B is a hallmark protein of the autophagy process, binding to the inner membrane of autophagosomes. When CQ hinders the function of autolysosomes, the contents of autolysosomes, including LC3B proteins present on the inner membrane of autophagosomes, cannot be degraded. Consequently, treatment with CQ leads to the accumulation of LC3B in cells, resulting in an increase in the levels of LC3B protein.

We hope this explanation addresses your concerns.

13) Figure 3 D isn't that useful, because the information is already provided in Figure 3 F.

Thank you for your suggestion. In Fig 3D, we investigated the promoting effect of overexpressing Linc00673-V3 on cellular autophagy, while in Fig 3F, we first knocked down the overall expression level of Linc00673 and then performed rescue experiments to eliminate the influence of different isoforms on cellular autophagy levels. Therefore, the two figures have a logical progression, and based on this, we believe that retaining Fig 3D is more appropriate.

14) Figure 3 F: The experimental data from the over-expression of the different isoforms alone (controls), are missing.

Thank you for your suggestion. As mentioned earlier, in Fig 3F, we did not directly compare the effects of overexpressing different isoforms on autophagy. We first knocked down the overall expression level of Linc00673 and observed a decrease in autophagy levels. Subsequently, we performed rescue experiments to verify whether Linc00673-V3 predominantly mediates this effect. Therefore, we did not compare the experimental results of overexpressing individual isoforms separately.

15) Please provide protein quantification for Figure 5 G/H

Thank you for your suggestion. We have added the densitometry graph for Fig. 5G/H to supplementary Figure 5C.

16) Figure 6 F/G: Over-expression of STUB1 results in lower Ub-Smad3 expression, while Linc00673-V3 over-expression reveals higher Ub-Smad3. This is in contrast to what the authors claim.

Thank you for your suggestion. Following your advice, we have checked Figure 6 F/G. There are a total of 5 lanes, including the IgG group. The results of overexpressing STUB1 are shown in the fourth lane, where the level of Ub-Smad3 is increased. Additionally, the results of co-overexpressing STUB1 and Linc00673-V3 are shown in the fifth lane, where the level of Ub-Smad3 is decreased. This is consistent with what we described in the manuscript.

17) Figure 8 K/L: The comparison between CDDP/ASO and ASO is lacking.

Thank you for your suggestion. We have supplemented the comparison between CDDP/ASO and ASO in Figure 8 K/L.

18) In the discussion section the authors need to discuss more thoroughly their findings on the basis of the current status of knowledge regarding the role of lncRNAs in autophagy in cancer in general and in NSCLC specifically.

Thank you for your suggestion. We have supplemented the discussion section regarding the relationship between lncRNAs and autophagy (Line478-495).

Reference

- 1 Lamark T, Svenning S, Johansen T. Regulation of selective autophagy: the p62/SQSTM1 paradigm. *Essays Biochem.* 2017; 61: 609-624.
- 2 Vargas JNS, Hamasaki M, Kawabata T, Youle RJ, Yoshimori T. The mechanisms and roles of selective autophagy in mammals. *Nat Rev Mol Cell Biol.* 2023; 24: 167-185.
- 3 Zhang JY, Zhang F, Hong CQ, et al. Critical protein GAPDH and its regulatory mechanisms in cancer cells. *Cancer Biol Med.* 2015; 12: 10-22.
- 4 Wang J, Yu X, Cao X, et al. GAPDH: A common housekeeping gene with an oncogenic role in pan-cancer. *Comput Struct Biotechnol J.* 2023; 21: 4056-4069.
- 5 Luo Y, Zheng S, Wu Q, et al. Long noncoding RNA (lncRNA) EIF3J-DT induces chemoresistance of gastric cancer via autophagy activation. *Autophagy.* 2021; 17: 4083-4101.
- 6 Zhu Y, Huang S, Chen S, et al. SOX2 promotes chemoresistance, cancer stem cells properties, and epithelial-mesenchymal transition by beta-catenin and Beclin1/autophagy signaling in colorectal cancer. *Cell Death Dis.* 2021; 12: 449.
- 7 Zhang F, Wang H, Yu J, et al. LncRNA CRNDE attenuates chemoresistance in gastric cancer via SRSF6-regulated alternative splicing of PICALM. *Mol Cancer.* 2021; 20: 6.
- 8 Guan H, Zhu T, Wu S, et al. Long noncoding RNA LINC00673-v4 promotes aggressiveness of lung adenocarcinoma via activating WNT/beta-catenin signaling. *Proc Natl Acad Sci U S A.* 2019; 116: 14019-14028.
- 9 Shi Q, Li Y, Li S, et al. LncRNA DILA1 inhibits Cyclin D1 degradation and contributes to tamoxifen resistance in breast cancer. *Nat Commun.* 2020; 11: 5513.
- 10 Wen S, Wei Y, Zen C, Xiong W, Niu Y, Zhao Y. Long non-coding RNA NEAT1 promotes bone metastasis of prostate cancer through N6-methyladenosine. *Mol Cancer.* 2020; 19: 171.
- 11 Chen S, Yang C, Wang ZW, et al. CLK1/SRSF5 pathway induces aberrant exon skipping of METTL14 and Cyclin L2 and promotes growth and metastasis of pancreatic cancer. *J Hematol Oncol.* 2021; 14: 60.
- 12 Curzer HJ, Perry G, Wallace MC, Perry D. The Three Rs of Animal Research: What they Mean for the Institutional Animal Care and Use Committee and Why. *Sci Eng Ethics.* 2016; 22: 549-565.
- 13 Lu W, Zhang H, Niu Y, et al. Long non-coding RNA linc00673 regulated non-small cell lung cancer proliferation, migration, invasion and epithelial mesenchymal transition by sponging miR-150-5p. *Mol Cancer.* 2017; 16: 118.

- 14 Mizushima N, Komatsu M. Autophagy: renovation of cells and tissues. *Cell*. 2011; 147: 728-741.
- 15 Klionsky DJ, Abdalla FC, Abeliovich H, et al. Guidelines for the use and interpretation of assays for monitoring autophagy. *Autophagy*. 2012;8(4):445-544. doi:10.4161/auto.19496.

March 7, 2024

RE: Life Science Alliance Manuscript #LSA-2023-02408-TR

Prof. Dajing Xia
Zhejiang University
School of Public Health
Zijingang Campus of Zhejiang University, Yuhangtang Road No.388, Zhejiang Province, P.R.China
Hang Zhou 310063
China

Dear Dr. Xia,

Thank you for submitting your revised manuscript entitled "Linc00673-V3 positively regulates autophagy by promoting Smad3 mediated LC3B transcription in NSCLC". We would be happy to publish your paper in Life Science Alliance pending final revisions necessary to meet our formatting guidelines.

- please be sure that the authorship listing and order is correct
- please remove the supplementary file. All supplementary figures should be uploaded separately, and their legends should be provided in the manuscript file
- please add ORCID ID for the corresponding (and secondary corresponding) author -- you should have received instructions on how to do so
- please add the Twitter handle of your host institute/organization as well as your own or/and one of the authors in our system
- please remove Graphical Abstract from the manuscript file and upload it separately with the file designation "Graphical Abstract"
- please note that the Graphical Abstract cannot also be a figure already used in the manuscript file. If you want to use Figure 9 as a Graphical Abstract, please remove it from the legend and callout. Otherwise, remove the Graphical Abstract
- please add callouts for Figure S8A-B and Table S5 to your main manuscript text

Figure Checks:

- please add sizes next to all blots
- please provide the original blots used to generate Figure 3D, and upload this as a Source Data file

A. FINAL FILES:

B. MANUSCRIPT ORGANIZATION AND FORMATTING:

Sincerely,

Reviewer #1 (Comments to the Authors (Required)):

The authors addressed all my concerns. I do not have any further questions.

Reviewer #2 (Comments to the Authors (Required)):

The authors have adequately addressed my concerns. I still wouldn't have used pcDNA3 for the overexpression experiments and used a plasmid with a promoter that isn't as strong but this isn't a deal breaker.

Reviewer #3 (Comments to the Authors (Required)):

The authors have addressed the recommendations we had proposed, and they included the majority of them in the manuscript.

March 13, 2024

RE: Life Science Alliance Manuscript #LSA-2023-02408-TRR

Prof. Dajing Xia
Zhejiang University
School of Public Health
Zijingang Campus of Zhejiang University, Yuhangtang Road No.388, Zhejiang Province, P.R.China
Hang Zhou 310063
China

Dear Dr. Xia,

Thank you for submitting your Research Article entitled "Linc00673-V3 positively regulates autophagy by promoting Smad3 mediated LC3B transcription in NSCLC". It is a pleasure to let you know that your manuscript is now accepted for publication in Life Science Alliance. Congratulations on this interesting work.

DISTRIBUTION OF MATERIALS:

Again, congratulations on a very nice paper. I hope you found the review process to be constructive and are pleased with how the manuscript was handled editorially. We look forward to future exciting submissions from your lab.

Sincerely,
